# Global Plane Waves from Local Gaussians: Periodic Charge Densities in a Blink

Jonas Elsborg [1 2]   Felix Ærtebjerg [* 1]   Luca Thiede [* 3 4]
Alán Aspuru-Guzik [3 4 5]   Tejs Vegge [1 2]   Arghya Bhowmik [1 2]

## Abstract

We introduce ELECTRAFI, a fast, end-to-end differentiable model for predicting periodic charge densities in crystalline materials. ELECTRAFI constructs anisotropic Gaussians in real space and exploits their closed-form Fourier transforms to analytically evaluate plane-wave coefficients via the Poisson summation formula. This formulation delegates non-local and periodic behavior to analytic transforms, enabling reconstruction of the full periodic charge density with a single inverse FFT. By avoiding explicit real-space grid probing, periodic image summation, and spherical harmonic expansions, ELECTRAFI matches or exceeds state-of-the-art accuracy across periodic benchmarks while being up to $633\times$ faster than the strongest competing method, reconstructing crystal charge densities in a fraction of a second. When used to initialize DFT calculations, ELECTRAFI reduces total DFT compute cost by up to $\sim 20\%$, whereas slower charge density models negate savings due to high inference times. Our results show that accuracy and inference cost jointly determine end-to-end DFT speedups, and motivate our focus on efficiency.

## 1. Introduction

Kohn-Sham density-functional theory (DFT) is the standard workhorse for electronic-structure calculations in physics, chemistry, and materials science (Hohenberg & Kohn, 1964; Kohn & Sham, 1965; Kresse & Furthmüller, 1996; Giannozzi et al., 2009; Jain et al., 2013). Despite its favorable accuracy-cost trade-off, canonical DFT remains expensive for large systems, long molecular-dynamics runs, and high-throughput screening. Most practical calculations are solved by self-consistent field (SCF) iteration: starting from an initial charge density, the code builds an effective potential, solves the Kohn-Sham eigenvalue problem, reconstructs a new density from the occupied orbitals, and repeats until convergence. Since this requires repeated diagonalization or related iterative linear algebra, even modest reductions in SCF iterations can translate into substantial wall-time savings. Machine learning methods for accelerating DFT broadly follow two strategies. Interatomic potentials directly predict observables such as energies, forces, or band gaps, bypassing the SCF cycle but no longer guaranteeing reproduction of a chosen DFT functional (Deringer et al., 2021; Batzner et al., 2022). Alternatively, one can keep the DFT calculation intact and improve its initialization by predicting the starting charge density (Brockherde et al., 2017). This is especially natural for plane-wave DFT codes, where the charge density is represented either as a periodic real-space grid over the unit cell or as Fourier coefficients over reciprocal-lattice vectors (Kresse & Furthmüller, 1996; Giannozzi et al., 2009; Payne et al., 1992). An ML-predicted density can therefore be inserted directly as the SCF initial guess. With this strategy, the standard DFT code still converges the final self-consistent result, but may require fewer iterations. Recent ML density models achieve strong accuracy for crystalline systems (Jørgensen & Bhowmik, 2022; Koker et al., 2024), but many evaluate the density by querying a neural network on a dense real-space grid. For large periodic systems, this inference cost can offset the saved SCF time. Thus, ML-initialized DFT requires not only accurate densities, but also fast inference. A gap therefore remains for a charge density model that is periodic, compatible with plane-wave DFT, captures long-range structure, and yields end-to-end wall-time reductions.

---

[*]Equal contribution   [1]Department of Energy Conversion and Storage, Technical University of Denmark, DK 2800 Kgs. Lyngby, Denmark [2]CAPeX Pioneer Center for Accelerating P2X Materials Discovery, DK 2800 Kgs. Lyngby, Denmark [3]Department of Computer Science, University of Toronto, Toronto, Ontario, Canada [4]Vector Institute for Artificial Intelligence, W1140-108 College Street, Toronto, ON M5G 0C6, Canada [5]Canadian Institute for Advanced Research (CIFAR), 661 University Ave., Suite 505, Toronto, ON M5G 1M1, Canada. Correspondence to: Jonas Elsborg <jels@dtu.dk>, Arghya Bhowmik <arbh@dtu.dk>.

*Proceedings of the $43^{rd}$ International Conference on Machine Learning*, Seoul, South Korea. PMLR 306, 2026. Copyright 2026 by the author(s).

**Conflict of Interest Disclosure.** The authors declare no financial or other substantive conflicts of interest that could reasonably be perceived to influence the work.

## 2. Related work & motivation

In contrast to MLIPs, density-based approaches or slight variations of their architectures can predict spatially resolved quantities, which are of great practical interest to understand the chemistry of a system and make targeted interventions. For example, Fukui functions (Parr & Yang, 1984) $f(\mathbf{r}) = \left(\frac{\partial \rho(\mathbf{r})}{\partial N}\right)_v$ are fundamental to predicting the most likely regions of a molecular system to be involved in a reaction. Other examples include the Quantum Theory of Atoms in Molecules (QTAIM), which allows the location and classification of bonds by calculating bond critical points of $\rho(\mathbf{r})$, again important to elucidate reaction mechanisms. The density can also be used directly in density-based generative models for drug design (Ragoza et al., 2022; Wang et al., 2022b), or as descriptors of electrochemical systems (Laubach et al., 2009; de Blasio et al., 2023; Shang et al., 2022) and electrocatalysts (Zheng et al., 2014; Koch et al., 2021). Densities can also be used to build cheap surrogates for observables like ionic diffusion (Kahle et al., 2018). However, the most obvious benefit is the potential for a significant reduction in wall time requirements associated with routine DFT calculations, since DFT is one of the largest global consumers of supercomputing time in chemistry and materials science (Gavini et al., 2023; Heinen et al., 2020). This demand is unlikely to diminish, particularly for complex interfaces and nanoscale systems (Hörmann et al., 2025). Both task-specific machine learning potentials and emerging foundation models for materials discovery are data-hungry and rely on DFT (or beyond-DFT) calculations for pretraining and continual improvement (Kulichenko et al., 2024; Pyzer-Knapp et al., 2025). Using ML-predicted charge densities as high-quality initial guesses for self-consistent DFT calculations offers a direct mechanism to reduce the number of electronic iterations required for convergence, thereby lowering the aggregate CPU/GPU hours consumed by large-scale electronic structure campaigns. Machine learning models for charge density prediction mirror two paradigms from electronic structure theory, which in the literature have been referred to as *orbital models* and *probe models*.

**Orbital models.** These approaches expand the density in atom-centered basis functions,

$$\rho(\boldsymbol{r}) = \sum_{i=1}^{N} \sum_{j=1}^{N_b^i} \sum_{m=-l_{i,j}}^{l_{i,j}} c_{i,j,m} \, \Phi_{\alpha_{i,j}, \, l_{i,j}, \, m, \mathbf{r_i}}(\boldsymbol{r}), \quad (1)$$

where $i$ indexes atoms, $j$ enumerates basis functions on atom $i$, and $m$ runs over magnetic sublevels. The coefficients $\{c_{i,j,m}\}$ weigh the basis functions $\Phi_{l,m}(\boldsymbol{r}) = R_l(r) Y_{lm}(\theta, \phi)$, which are products of radial basis functions $R_l(r)$ and a spherical harmonic $Y_{lm}(\theta, \phi)$. Learning to predict the density can thus be framed as the prediction of

the coefficients $\{c_{i,j,m}\}$ and, in some cases, refining parts of the radial function (Fabrizio et al., 2019; Qiao et al., 2022; Rackers et al., 2023; Cheng & Peng, 2024; del Rio et al., 2023; Febrer et al., 2025; Fu et al., 2024). However, a fixed atom-centered basis generally limits expressivity, often forcing large expansions to capture inter-atomic features. Bond-centered augmentations increase accuracy (Fu et al., 2024), but learned, per-atom displacements provide a more general solution (Elsborg et al., 2026): simple 3D Gaussians placed at predicted offsets deliver high accuracy while eliminating high-order spherical harmonics. This leads to much faster inference and avoids introducing a strong model dependence on the choice of basis sets and their associated coefficient representation. Current orbital-based models perform well on small molecules and organic chemistries, but in solids, the non-local features of the charge density require large orbital expansions with many diffuse and high-angular-momentum functions, which undermines the efficiency of the LCAO ansatz for materials (Lejaeghere et al., 2016). As a result, high accuracy has not been demonstrated in general for inorganic crystals and metals (Cheng & Peng, 2024; Kim & Ahn, 2024; Fu et al., 2024).

**Probe models.** Another class of models treat points of the numerical grid (Cerjan, 2013) as nodes of a big graph neural network and allows each point to receive messages from an atomic graph encoder (Gong et al., 2019; Jørgensen & Bhowmik, 2022; Koker et al., 2024; Pope & Jacobs, 2024; Li et al., 2025). This yields high functional flexibility and significant reductions in the number of SCF steps when utilized as initial guesses in regular DFT calculations (Koker et al., 2024). However, sending messages to all grid points requires many GNN queries and, therefore, higher inference time compared to orbital approaches (Fu et al., 2024). Common to most modern density prediction models is that they build on graph neural network models and/or inter-atomic potentials, which we describe briefly below.

**Molecular modeling with graph neural networks** Graph Neural Networks (GNNs) provide a powerful framework for processing data that can be structured as graphs. Formally, a graph is defined as a pair $\mathcal{G} = (\mathcal{V}, \mathcal{E})$, where $\{V\}$ denotes a set of vertices (nodes) and $\mathcal{E} \subseteq \mathcal{V} \times \mathcal{V}$ the edges connecting them. Information is represented as a set of feature vectors $\{\mathbf{h}_i\}$ attached to each vertex $i \in \mathcal{V}$. GNNs process this data through a series of message-passing and pooling layers that update node features by aggregating information from neighboring nodes while maintaining permutation equivariance, i.e., the model's output must be independent of the arbitrary ordering of the nodes in the input. Following a stack of these layers, a per-node or global readout yields the final output.

When applied to molecular systems, molecules are modeled

as graphs embedded in three-dimensional space. In this representation, each node $i$ is associated with a Cartesian atomic position $\mathbf{r}_i = (x_i, y_i, z_i)$. To capture the geometric structure of the molecule, message-passing functions are designed to depend explicitly on interatomic distance vectors $\mathbf{r}_{ij} = \mathbf{r}_i - \mathbf{r}_j$. This allows the network to incorporate the local environment of atoms, which determines chemical behavior. The development of atomistic GNN architectures has been largely driven by Machine Learning Interatomic Potentials (MLIPs) that predict energies and forces based on the molecular geometry. MLIPs can be broadly classified into two categories based on their treatment of physical symmetries:

1. Symmetry-constrained: These architectures either utilize features that are invariant (distances, angles, etc.) or transform predictably (equivariantly) under rotations and reflections of the molecule. For example, SchNet uses only distances (Schütt et al., 2017), DimeNet, and ANI (Smith et al., 2017) also include angles (Gasteiger et al., 2020), and QuiNet (Wang et al., 2023) derives a way for efficient construction of higher-order invariant features. The first equivariant model was Tensor Field Networks (Thomas et al., 2018), which was further improved, for example, by MACE (Batatia et al., 2022), Equiformer (Liao et al.) and NequIP (Batzner et al., 2022).

2. Unconstrained Models: Models like EScAIP (Qu & Krishnapriyan, 2024) do not explicitly enforce geometric symmetries within the architecture's internal layers, prioritizing flexibility and computational efficiency instead.

There is a fundamental trade-off between these approaches: equivariant and invariant models are typically more data-efficient, since they need not learn rotational invariance from scratch, but their added architectural complexity increases computational cost. Unconstrained models, by contrast, are usually faster and better suited for large-scale simulations when sufficient training data is available.

Modified GNN architectures have also recently been used to predict the Hamiltonian and density matrices (Li et al., 2022; Febrer et al., 2025), virial tensor (Wang et al., 2018), Hessian matrix (Burger et al., 2025), or dipole moments (Unke & Meuwly, 2019).

**Frequency-domain models.** Some recent models have used spectral operators in Fourier/reciprocal space to encode periodicity and long-range physics directly in $k$-space, e.g., neural operators parameterized in the Fourier domain and reciprocal-space augmentations for long-range interactions (Li et al., 2021; Tran et al., 2023; Kosmala et al.,

2023). Such approaches have been effective across periodic systems and lattices such as metasurfaces, composites, and sonic crystals, where FFT-based layers provide global, low-$k$ coupling (Rashid et al., 2022; Kang et al., 2025; Wagner et al., 2024). A related line of work augments short-range geometric GNNs with a mesh-based, FFT-accelerated long-range channel that learns a Fourier-domain influence kernel and exchanges information bidirectionally between atoms and mesh at each layer (Wang et al., 2024). Fixed Ewald terms can also be replaced with a learned Fourier domain convolution, where per-atom latent variables feed a sum-of-Gaussians spectral multiplier evaluated with FFTs to capture diverse long-range decays at near-linear cost (Ji et al., 2025). For electron density specifically, plane-wave operator heads have been used as global, lattice-aligned corrections on top of atom-centered orbitals, though in practice they offer only minimal incremental benefit over strong real-space/orbital backbones (Kim & Ahn, 2024). Thus, while reciprocal space is the natural domain for periodic systems, prior frequency-domain add-ons for ML charge density modeling have demonstrated little benefit.

**Contributions.** We introduce the Electronic Tensor Reconstruction Algorithm with Fourier Inversion (ELECTRAFI), a fast and accurate model for charge density prediction in periodic inorganic materials. Our main contributions are:

- We replace dense neural evaluation of every grid point with a local-to-global density representation based on floating Gaussians. This admits a closed-form analytic Fourier transformation that enables us to construct an efficient periodization using the Poisson summation formula, without periodic image summation.

- We demonstrate state-of-the-art accuracy on several periodic benchmarks while achieving sub-second inference and up to $633\times$ speedup compared to the strongest prior grid-based density model.

- We demonstrate that ELECTRAFI is the first model to achieve a consistent reduction in end-to-end DFT computation time when using machine-learned charge densities as SCF initial guesses in materials.

## 3. Methods

For clarity, we summarize the main notation used throughout the paper in Appendix A.1.

**ELECTRAFI.** A mixture-of-Gaussians ansatz has been established as an efficient representation for molecular

charge densities (Elsborg et al., 2026):

$$\rho(\mathbf{r}) \;=\; \sum_{j=1}^{N_{\mathcal{N}}} w^{(j)} \, \mathcal{N}(\mathbf{r}\,;\,\boldsymbol{\mu}^{(j)}, \boldsymbol{\Sigma}^{(j)}), \qquad (2)$$

where

$$\mathcal{N}(\mathbf{r};\mu,\Sigma) \;=\; \frac{\exp\!\left[-\tfrac{1}{2}(\mathbf{r}-\mu)^{\top}\Sigma^{-1}(\mathbf{r}-\mu)\right]}{(2\pi)^{3/2}\sqrt{\det\Sigma}}, \quad (3)$$

with centers $\boldsymbol{\mu}^{(j)} \in \mathbb{R}^3$, positive-definite covariances $\boldsymbol{\Sigma}^{(j)} \in \mathbb{R}^{3\times3}$, and signed weights $w^{(j)} \in \mathbb{R}$ (see Figure 1).

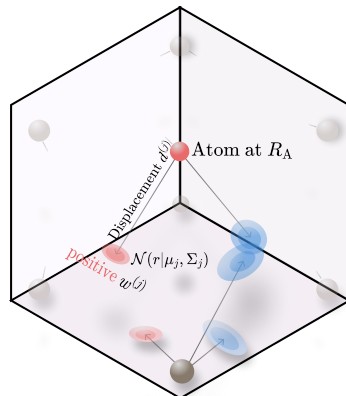

*Figure 1.* The electron density is parametrized by a set of Gaussians, where each component $j$ has a signed weight $w^{(j)}$, displacement from anchor $\mathbf{d}^{(j)}$, and covariance $\boldsymbol{\Sigma}^{(j)}$.

However, in periodic electronic structure theory, charge densities are most often represented via plane-wave expansion coefficients in reciprocal space, which ensures periodicity by construction (See Appendix A.2 for details on plane-wave representations) (Payne et al., 1992). From Bloch's theorem (Bloch, 1929), we know that the electron density of a periodic system with periodicity defined by the lattice $\Lambda = \{\mathbf{R} \in \mathbb{R}^3 \mid \mathbf{R} = n_1\mathbf{a}_1 + n_2\mathbf{a}_2 + n_3\mathbf{a}_3, \quad n_i \in \mathbb{Z}\}$ has to be periodic and smooth. We could simply form the periodic continuation $\rho_{\text{per}}(\mathbf{r}) = \rho(\mathbf{r} - \mathbf{R}(\mathbf{r}))$, where $\mathbf{R}(\mathbf{r})$ is the unique lattice vector that shifts $\mathbf{r}$ back into the unit cell. This can be thought of as replicating the unit cell across space. However, for general $\rho$, this leads to discontinuities on the boundaries of the unit cells. To enforce smoothness, we could periodize Equation (2) by summing Gaussian contributions from all periodic images:

$$\rho_{\mathbf{P}}(\mathbf{r}) = \sum_{\mathbf{R} \in \Lambda} \rho(\mathbf{r} + \mathbf{R}) \qquad (4)$$

Clearly, we cannot sum over infinitely many lattice vectors. Instead, a straightforward, practically implementable approximation is to only sum over a finite number of neighbor images:

$$\rho_{\mathbf{P}'}(\mathbf{r}) = \sum_{\mathbf{R} \in \Lambda_N} \rho(\mathbf{r} + \mathbf{R}) \qquad (5)$$

where $\Lambda_N = \{\mathbf{R} \in \mathbb{R}^3 \mid \mathbf{R} = n_1\mathbf{a}_1 + n_2\mathbf{a}_2 + n_3\mathbf{a}_3, \quad n_i \in \{-N, ..., N\}\}$. For a model using floating Gaussian orbitals, this would mean each grid point "sees" Gaussians from up to $N$ periods/primitive cells away. However, with this approach, the resulting function is not actually periodic, and a periodic continuation still has discontinuities that only decay if we sum over enough images. This is very expensive, though; even for $N = 1$ the evaluation already becomes $27\times$ as expensive as the evaluation of a single unit cell.

An alternative approach that can periodize any Schwartz function $\rho$ (a function is a Schwartz function if its value and all its derivatives decay to zero at infinity) is using Poisson's summation formula, given by

$$\sum_{\mathbf{R} \in \Lambda} \rho(\mathbf{r} + \mathbf{R}) = \frac{1}{|\Omega|} \sum_{\mathbf{G} \in \Lambda^*} \hat{\rho}(\mathbf{G}) e^{i\mathbf{G}\cdot\mathbf{r}} \qquad (6)$$

with reciprocal lattice $\Lambda^*$, simulation cell $\Omega$, cell volume $|\Omega|$, and Fourier transform $\hat{\rho}$. In practice, we also have to truncate the reciprocal lattice $\Lambda^*$, but the RHS of Equation (6) remains a periodic function even with a truncated $\Lambda^*$. Instead, truncation only acts as a low-pass filter, which is much more acceptable compared to discontinuities, as we know that the DFT charge densities are, in practice, also low-passed functions.

Therefore, our model, ELECTRAFI, represents the valence charge density using a finite set of anisotropic Gaussians that are periodized using Poisson's summation formula. Conceptually, the Gaussian parameters are predicted in real space, but the density itself is constructed entirely in reciprocal space. For clarity, we introduce the auxiliary non-periodic function

$$\tilde{\rho}(\mathbf{r}) \;=\; \sum_{j=1}^{N_{\mathcal{N}}} w^{(j)} \, \mathcal{N}(\mathbf{r}\,;\,\boldsymbol{\mu}^{(j)}, \boldsymbol{\Sigma}^{(j)}), \qquad (7)$$

where all parameters $w^{(j)}$, $\mu^{(j)}$ and $\boldsymbol{\Sigma}^{(j)}$ are predicted by ELECTRAFI.

Unlike prior density models based on floating Gaussians (Elsborg et al., 2026), the function $\tilde{\rho}(\mathbf{r})$ is not interpreted as the physical real-space charge density, nor is it ever evaluated or discretized in real space. Instead, we exploit the self-reciprocity of Gaussians under Fourier transformation, whereby a Gaussian maps to a Gaussian in reciprocal space and admits a closed-form analytic expression,

$$\begin{aligned} \mathcal{F}[\mathcal{N}(\mathbf{r};\mu,\Sigma)]\,(\mathbf{G}) &\;=\; \int_{\mathbb{R}^3} \mathcal{N}(\mathbf{r};\mu,\Sigma)\, e^{-i\mathbf{G}\cdot\mathbf{r}}\, d\mathbf{r} \\ &\;=\; \exp\!\left[-\tfrac{1}{2}\,\mathbf{G}^{\top}\Sigma\mathbf{G}\right] e^{-i\mathbf{G}\cdot\mu}. \end{aligned} \qquad (8)$$

The Fourier transformation $\hat{\rho}$ of (7) is therefore:

$$\hat{\rho}(\mathbf{G}) \;=\; \sum_{j=1}^{N_{\mathcal{N}}} w^{(j)} \exp\!\left[-\tfrac{1}{2}\,\mathbf{G}^{\top}\boldsymbol{\Sigma}^{(j)}\mathbf{G}\right] e^{-i\,\mathbf{G}\cdot\boldsymbol{\mu}^{(j)}}, \quad (9)$$

Applying an inverse fast Fourier transform (IFFT) to $\widehat{\rho}$ essentially evaluates the Poisson summation formula (6) and recovers the periodic, smooth real-space density $\rho(\mathbf{r})$:

$$\rho(\mathbf{r}) = \text{IFFT}(\hat{\rho}(\mathbf{G}))\,(\mathbf{r}), \qquad \rho(\mathbf{r}) \in \mathbb{R}. \qquad (10)$$

Note that the closed-form Fourier transformation was crucial in our construction, as it avoids expensive and noisy numerical Fourier transformations that would be necessary for a general functional form. Consequently, unlike other density models (Jørgensen & Bhowmik, 2022; Koker et al., 2024), ELECTRAFI eliminates the computational overhead associated with explicitly handling periodic images in real-space density evaluation. To ensure that the output density sums to the correct number of valence electrons, we rescale all weights $\{w^{(j)}\}$ by a single positive factor so that

$$\int_{\Omega} \rho(\mathbf{r})\,d\mathbf{r} = \sum_{\mathbf{r}} \rho(\mathbf{r})\Delta v = N_e, \qquad (11)$$

where $\Omega \subset \mathbb{R}^3$ is the simulation cell, and $\Delta v$ is the voxel volume.

Finally, it is natural to ask whether the plane-wave coefficients $\hat{\rho}(\mathbf{G})$ could be predicted directly. This would require learning a global, resolution-dependent readout whose output dimension scales with the FFT grid and plane-wave cutoff, rather than with the number of local atomic degrees of freedom. Prior work has explored such a direction (Kim & Ahn, 2024), where a plane-wave branch yields only a marginal improvement when added to local Gaussian-type orbital features, and is highly inaccurate as a standalone representation ansatz at practical basis sizes. In contrast, ELECTRAFI predicts local Gaussian parameters and obtains the global reciprocal-space coefficients analytically through the closed-form Fourier transform.

> ELECTRAFI leverages the Poisson summation formula and closed-form Fourier transform of Gaussians to efficiently represent periodic charge densities without explicit periodic image summation.

**Gaussian indexing and per-atom allocation.** Given a periodic molecular graph, ELECTRAFI outputs per-atom channels

$$S \in \mathbb{R}^{N \times C}, \quad V \in \mathbb{R}^{N \times C \times 3}, \quad T \in \mathbb{R}^{N \times C \times 3 \times 3},$$

which parameterize Gaussian amplitudes, directions, and shapes. Here, $N$ is the number of atoms and $C$ the channel width. We assign a fixed number $M$ of Gaussians per valence electron: for atom $a$ with valence $v_a$, this yields $n_a = M v_a$ Gaussians. The total number of Gaussian components in system $s$ is therefore

$$N_{\mathcal{N}} = M \sum_{a \in s} v_a. \qquad (12)$$

For each atom, the first $n_a$ of its $C$ channels are used to construct Gaussians (with $C \geq \max_a n_a$), while the remaining channels are unused. We index Gaussians globally by concatenating per-atom slots, $j \equiv (a, c)$ with $c \in \{1, \dots, n_a\}$, and write $\boldsymbol{\mu}^{(j)}, \boldsymbol{\Sigma}^{(j)}$, and $w^{(j)}$ for the parameters of Gaussian $\mathcal{N}^{(j)}$. To ensure consistency with the reference data, we use the same element-specific valence conventions used in dataset generation. See Appendix B.3 for more details.

**Gaussian parametrization.** Each Gaussian $\mathcal{N}^{(j)}$ is parametrized by the set $(w^{(j)}, \boldsymbol{\mu}^{(j)}, \boldsymbol{\Sigma}^{(j)})$, specifying its weight, location and covariance matrix.

*Weights.* We produce $w^{(j)}$ from $S^{(j)}$ via a multi-layer perceptron (MLP) and a signed, magnitude-stable factorization:

$$\begin{aligned} w^{(j)} &= \tanh\!\big(s^{(j)}\big), \\ s^{(j)} &= f_w\big(S^{(j)}\big), \end{aligned} \qquad (13)$$

where $f_w$ is a MLP.

*Centers.* We place each Gaussian as a displacement from its host atom $a(j)$:

$$\boldsymbol{\mu}^{(j)} = \mathbf{R}_{a(j)} + \mathbf{d}^{(j)}, \qquad \mathbf{d}^{(j)} \in \mathbb{R}^3, \qquad (14)$$

with $\mathbf{d}^{(j)}$ drawn from the vector channel $V^{(j)}$.

*Covariances.* SPD covariances are parametrized through a Gram factorization,

$$\boldsymbol{\Sigma}^{(j)} = \gamma^{(j)} \mathbf{A}^{(j)} \mathbf{A}^{(j)\top} \qquad (15)$$

where $\mathbf{A}^{(j)} \in \mathbb{R}^{3 \times 3}$ comes from the tensor channel $T^{(j)}$ and $\gamma^{(j)}$ is a scaling factor predicted using a MLP on the scalar features $S^{(j)}$.

**Attention-based backbone.** We use a modified EScAIP backbone to update scalar node/edge streams via multi-head attention over PBC-aware neighborhoods, followed by position-wise feed-forward transformations (Qu & Krishnapriyan, 2024). This concentrates capacity in attention/FFNs and constructs geometry-aware vectors/tensors via dyadic reductions. It avoids expensive SO(3) tensor products and high-order irreps whose cost grows steeply with $L_{\max}$ and Gaunt contractions (Thomas et al., 2018; Batzner et al., 2022; Batatia et al., 2022; Liao et al.; Passaro & Zitnick, 2023). Compared to angle/dihedral-basis models (Gasteiger et al., 2020; 2021; 2022), we avoid explicit triplet enumeration. Our vector/tensor heads are masked, attention-weighted reductions over unit edge directions and their traceless dyads, keeping complexity linear in the number of edges. We construct dyadic neighbor aggregation layers which produce the per-atom channels $S, V, T$, which in turn are used to construct $\big(\mu^{(j)}, \Sigma^{(j)}, w^{(j)}\big)$ (details in Appendix B) which parametrize our Gaussians through Equations (13), (14), and (15).

The full end-to-end modeling workflow is shown in Figure 2.

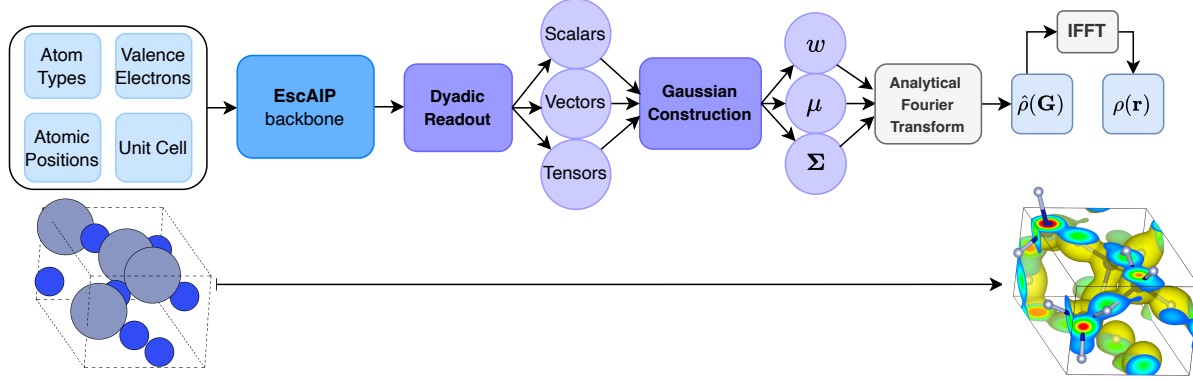

*Figure 2.* End-to-end ELECTRAFI workflow. Given the atomic structure, the attention-based backbone and custom dyadic readout layers predict the parameters of an intermediate real-space Gaussian mixture. The model then analytically Fourier transforms this mixture to obtain plane-wave coefficients, from which the periodic real-space charge density is reconstructed with a single inverse FFT.

**Training objective.** Following the evaluation convention of prior charge density models (Jørgensen & Bhowmik, 2022; Koker et al., 2024; Elsborg et al., 2026), we use normalized mean absolute error (NMAE) as the main density metric. ELECTRAFI is trained directly on this objective,

$$\mathcal{L} = \mathrm{NMAE}(\rho_{\mathrm{pred}}, \rho_{\mathrm{ref}}) = \frac{\int_\Omega |\rho_{\mathrm{ref}}(\mathbf{r}) - \rho_{\mathrm{pred}}(\mathbf{r})|\, dV}{\int_\Omega \rho_{\mathrm{ref}}(\mathbf{r})\, dV}. \tag{16}$$

Here, $\Omega$ denotes the periodic simulation cell. This loss is evaluated on the full real-space density grid for each structure, without grid subsampling. We report all density prediction errors as $\mathrm{NMAE}[\%] = 100 \times \mathrm{NMAE}$.

## 4. Experiments

Using the NMAE metric defined in Equation (16), we train ELECTRAFI on a suite of periodic charge density benchmarks. We note that ChargE3Net is substantially more accurate on periodic materials than other published models (see also Table 1), and Koker et al. (2024) is the only work to report comprehensive evaluations on generalized DFT cost reduction experiments for such systems. A central motivation of this work is therefore to assess whether ELECTRAFI can reach comparable accuracy, while avoiding ChargE3Net's computational bottlenecks. The main ELECTRAFI model is trained on the full Materials Project (MP) distribution (MP-Full), which we split into 117,876 training structures, 512 validation structures and 2,000 test structures. The split is similar but not identical to Koker et al. (2024). Achieving speed and scalability on this dataset is challenging: the largest charge density grid files exceed 4 GB, and inference must produce up to 157 million floats per material. In addition, we evaluate the main model on 2,000 structures from the GNoME dataset (Merchant et al., 2023) with recomputed VASP charge densities, probing generalization beyond the MP dataset. For these two datasets, we

also run our own experiments with the released ChargE3Net model. Both ELECTRAFI and ChargE3Net were evaluated on a single NVIDIA A100-40GB GPU, while parallelized VASP DFT calculations were performed on 24 CPU cores, specifically Intel Xeon E5-2650 2.20GHz CPUs of the Broadwell generation, using 256 GB of RAM memory.

**Benchmark results.** In Table 1, we report the NMAE [%] and average inference time $t_{inf}$ for ELECTRAFI and other recent models. On the MP-Full test set, ELECTRAFI achieves an NMAE [%] of $0.58\%$, which is slightly higher than ChargE3Net's error of $0.54\%$, but at a roughly $463\times$ faster average inference time of $0.17s$, whereas ChargE3Net takes $\approx 1.3$ minutes on average. On GNoME structures, ELECTRAFI has a modestly higher error ($0.93\%$ NMAE) compared to ChargE3Net ($0.69\%$ NMAE), but is again much faster, with average inference times that are roughly $302\times$ lower ($0.11s$ versus $33.28s$). To demonstrate the scaling behavior of both models as well as DFT on the test sets, Figure 3 shows how inference times scale with system size. Although canonical Kohn–Sham DFT exhibits cubic asymptotic scaling, the system sizes considered here lie below the regime where diagonalization dominates, resulting in an approximately linear scaling with system size for DFT computation times in VASP. However, the slope and intercept of ELECTRAFI still demonstrate superior scaling in both datasets. To understand the failure modes of each model, Appendix F contains a detailed element-wise error analysis that compares the average prediction error for ELECTRAFI and ChargE3Net across structures containing each element. It highlights a systematic element-dependent performance split between ELECTRAFI and ChargE3Net. Specifically, ELECTRAFI performs best for alkali/alkaline-earth elements, halogens, and heavy elements (in particular Ac), which typically appear in more ionic or delocalized bonding environments, while ChargE3Net is better for light

*Table 1.* Charge density prediction on test sets of periodic benchmarks. A dash (—) indicates not reported. Results reproduced from other work are indicated by superscript: [a] (Koker et al., 2024), [b] (Kim & Ahn, 2024), [c] (Fu et al., 2024), [d] (Chen et al., 2025).

| Dataset | Metric | ELECTRAFI | ChargE3Net | DeepDFT | SCDP | GPWNO | InfGCN |
|---|---|---|---|---|---|---|---|
| **MP-Full** | NMAE [%] | 0.58 | **0.54** | 0.80[a] | — | — | — |
| | $t_{inf}$ [s] | **0.17** | 78.73 | — | — | — | — |
| | Speedup vs ChargE3Net | **463×** | — | — | — | — | — |
| **GNoME** | NMAE [%] | 0.93 | **0.69** | — | — | — | — |
| | $t_{inf}$ [s] | **0.11** | 33.28 | — | — | — | — |
| | Speedup vs ChargE3Net | **302×** | — | — | — | — | — |
| **ECD-HSE06** | NMAE [%] | **1.35** | 1.53[d] | — | — | — | — |
| | $t_{inf}$ [s] | **0.05** | 31.65 | — | — | — | — |
| | Speedup vs ChargE3Net | **633×** | — | — | — | — | — |
| **MP-Mixed** | NMAE [%] | **1.24** | — | 11.50[b] | — | 4.83[b] | 5.11[b] |
| | $t_{inf}$ [s] | 0.17 | — | — | — | — | — |
| **Cubic** | NMAE [%] | **1.37** | — | 10.37[b] | 2.59[c] | 7.69[b] | 8.98[b] |
| | $t_{inf}$ [s] | 0.08 | — | — | — | — | — |

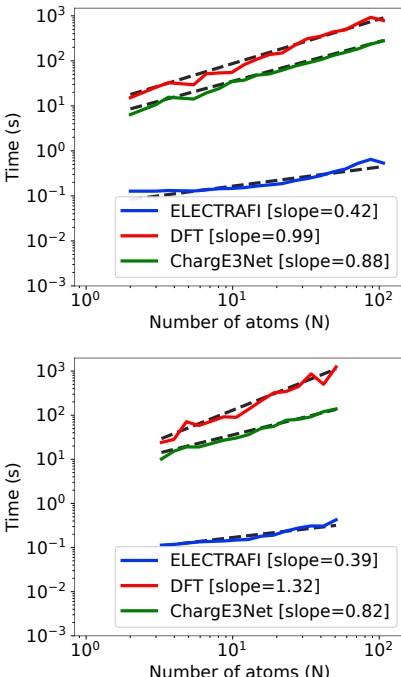

*Figure 3.* Logarithmic scaling plots showing wall-clock time versus system size for ELECTRAFI, DFT, and ChargE3Net. **Top:** Materials Project (MP) test set. **Bottom:** GNoME test set.

covalent elements and open-shell transition metals. The two models thus exhibit complementary inductive biases aligned with our expectations, since ELECTRAFI favors smoother, more globally distributed charge densities while ChargE3Net better captures localized, directional bonding features. Furthermore, in Appendix G, we show charge density visualizations for the best- and worst-performing test structures of ELECTRAFI on both the MP-Full and GNoME benchmarks, alongside the corresponding ChargE3Net pre-

dictions for the same materials. To enable direct comparison with prior work, we also train and evaluate ELECTRAFI on the MP-Mixed (Kim & Ahn, 2024) and Cubic (Wang et al., 2022a) benchmarks, achieving 1.24 % and 1.37 % NMAE, respectively. These results outperform all published models (Table 1) while maintaining extremely favorable inference times. The datasets described so far all use the PBE (Perdew et al., 1996) exchange–correlation functional. To assess performance on densities originating from higher-order functionals, we also train and test an ELECTRAFI model on the HSE06 (a hybrid functional) subset of the ECD dataset (Chen et al., 2025). When trained from scratch, ELECTRAFI attains state-of-the-art accuracy at 1.35 % NMAE, improving upon ChargE3Net (1.53%) while delivering an average 633× speedup (0.04 s vs. 31.65 s). Because ELECTRAFI does not use an explicitly rotation-equivariant architecture, we also use the ECD-HSE06 benchmark to test ELECTRAFIs sensitivity to arbitrary coordinate-frame rotations. The default DFT initialization setting uses structures in a fixed lattice frame, but rotation robustness is relevant for broader uses of ML-predicted densities as descriptors or intermediate representations. Without rotation augmentation, evaluating the ECD-HSE06 model on randomly rotated structures increases the test error from 1.35% to 1.69% NMAE. However, training with random rotations mitigates this loss and slightly improves the rotated-test error to 1.33% NMAE, indicating that ELECTRAFI can learn rotation-robust behavior through simple data augmentation. Full details are given in Appendix C.

We provide details on the size of the individual datasets as well as training settings for each dataset in Appendix E.

### 4.1. DFT acceleration experiments

We perform extensive evaluations of the ability of both ChargE3Net and ELECTRAFI to accelerate routine DFT

*Table 2.* Results from DFT calculations using various initialization methods for the filtered and recomputed non-magnetic subset of the test files from MP and GNoME. All numbers for time and steps saved are reported relative to the default SAD guess.

| Dataset | Metric | SAD (Default) | SC (Converged) | ELECTRAFI | ChargE3Net |
|---------|--------|---------------|----------------|-----------|------------|
| **MP** | NMAE ↓ | | | 0.55 % | 0.50 % |
| | ML Init Time ↓ | | | 0.17 s | 72.11 s |
| | SCF Steps ↓ | 16.80 | 8.73 | 13.33 | 12.09 |
| | DFT Time ↓ | 266.34 s | 161.98 s | 219.57 s | 208.26 s |
| | Total Time ↓ | 266.34 s | 161.98 s | 219.74 s | 280.37 s |
| | DFT Steps Saved ↑ | | 48.04 % | 20.65 % | 28.03 % |
| | DFT Time Saved ↑ | | 39.18 % | 17.56 % | 21.81 % |
| | Total Time Saved ↑ | | 39.18 % | 17.50 % | -5.27 % |
| **GNoME** | NMAE ↓ | | | 0.88 % | 0.59 % |
| | ML Init Time ↓ | | | 0.11 s | 28.29 s |
| | SCF Steps ↓ | 13.49 | 6.88 | 10.11 | 9.50 |
| | DFT Time ↓ | 119.98 s | 77.91 s | 95.06 s | 92.67 s |
| | Total Time ↓ | 119.98 s | 77.91 s | 95.17 s | 120.96 s |
| | DFT Steps Saved ↑ | | 51.73 % | 25.05 % | 29.58 % |
| | DFT Time Saved ↑ | | 39.75 % | 20.77 % | 22.76 % |
| | Total Time Saved ↑ | | 39.75 % | 20.68 % | -0.82 % |

calculations on the MP-Full and GNoME test sets. We initialize DFT calculations with predicted charge densities from both models and record the reduction in self-consistent field (SCF) steps and overall calculation time needed to converge the density and energies. We provide full details on these experiments in Appendix D. Following prior work on SCF acceleration in materials (Koker et al., 2024), we restrict these experiments to the non-magnetic subsets of the MP-Full and GNoME test sets. We recompute reference densities with VASP 5.4.4 using MP-style inputs and pseudopotentials, including the same Hubbard-$U$/LMAXMIX heuristics for GNoME. Neither ChargE3Net nor ELECTRAFI predicts augmentation occupancies, which were taken from the Materials Project dataset or computed self-consistently for the GNoME structures. Since ELECTRAFI enforces electron-count normalization by construction, we apply the same normalization to ChargE3Net to ensure a fair comparison. We display the results of our experiments in Table 2. In the table, we also include results using the converged self-consistent (SC) density as an initial guess, which can be interpreted as an upper bound on the achievable acceleration from improved density initialization. The results show that only ∼ 75–80% of SCF step reductions are realized as DFT wall time savings. Furthermore, when inference cost is accounted for, ChargE3Net leads to a negative percentage time saved, i.e., an overall *slowdown*. In contrast, ELEC-TRAFI achieves a significant net reduction in mean total compute time on both datasets because its inference time is nearly negligible compared with the DFT calculation. Figure 4 further shows that this advantage is maintained across system sizes, and ELECTRAFI is faster than the default SAD initialization on average, whereas ChargE3Net loses most or all of its DFT-only savings once inference time is included. However, even for ELECTRAFI there are individual structures that show negative net savings: total wall

time increases for 13.37% of MP-Full structures and 7.18% of GNoME structures, whereas ChargE3Net increases total wall time for 84.35% and 74.56% of structures, respectively. While hardware choices will affect wall times, the A100

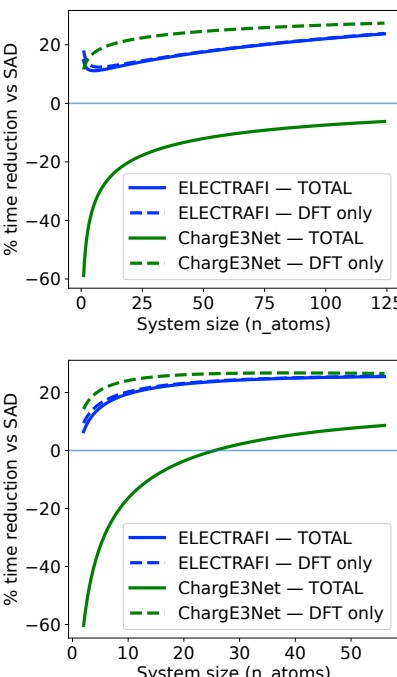

*Figure 4.* End-to-end versus DFT-only wall-time reduction as a function of system size. Structures are grouped by number of atoms, and curves show the smoothed mean percentage time reduction relative to the default SAD initialization. "TOTAL" includes both ML inference time and the subsequent DFT calculation, while "DFT only" excludes ML inference time and measures only the reduction in DFT wall time. Positive values indicate speedup relative to SAD, negative values indicate an overall slowdown. Top: Materials Project (MP-Full) test set. Bottom: GNoME test set.

GPU used for ML inference is more modern than the Broadwell generation of CPUs used for the VASP calculations. Faster CPUs would reduce the DFT computation time and thus the relative savings for both models in Table 2. In particular, for ChargE3Net, the high inference time would make the percentage slowdown even worse. In fact, Figure 4 shows that ChargE3Net does not reduce the net computation time on MP in any domain, while for GNoME, there is a slight average saving for system sizes above 30 atoms. When considering only the DFT wall time, ChargE3Net has a slight advantage over ELECTRAFI, but this diminishes as the system size grows, and on the GNoME dataset it completely disappears for systems with >50 atoms. One possible explanation is that SCF convergence is governed by how the initialization error projects onto slowly converging long-wavelength charge modes, rather than by global density error metrics such as NMAE.

> ELECTRAFI achieves up to $633\times$ speedup while matching or exceeding state-of-the-art accuracy compared to existing models, positioning it as the first model that achieves end-to-end time savings for DFT calculations of materials.

## 5. Discussion

**Accuracy, Cost, and SCF Reduction.** ELECTRAFI's accuracy-cost balance exceeds all prior models and enables higher iteration rates while broadening the range of applications for ML-predicted densities. However, practical considerations remain before large-scale deployment within conventional DFT workflows. Future work is needed to identify which training protocols best correlate with SCF savings, including the choice of loss function and grid sampling strategy. We observe that periodic charge densities exhibit higher error floors than molecular benchmarks such as QM9 (Koker et al., 2024; Fu et al., 2024; Elsborg et al., 2026). As a consequence, larger DFT speedups are achievable in molecular settings, where up to $\sim 50\%$ of SCF iterations can be eliminated (Elsborg et al., 2026). We attribute this difference mainly to the increased complexity of periodic electronic structures and the much larger chemical/elemental space encompassed. However, the analysis in Appendix G suggests that ELECTRAFI and ChargE3Net dominate complementary regimes, motivating hybrid models that combine efficient global Fourier-space representations with localized real-space corrections. Even more so since the computational cost of ChargE3Net can be reduced with better probing strategies or hardware optimization.

**Better Density Benchmarks Are Needed.** In practice, end-to-end reductions in computational cost are more relevant than grid-level accuracy alone, and charge density models should be evaluated in direct integration with DFT codes to understand their impact and ensure consistent treatment of pseudopotentials and valence electron definitions. This is currently limited by dataset shortcomings. While the Materials Project is the most widely used periodic charge density resource, it was created over a longer period with evolving best practices for DFT, resulting in occasional inconsistencies. Even newer datasets (Chen et al., 2025) lack sufficient metadata for reproducible end-to-end DFT experiments, and the absence of standardized benchmarks remains a bottleneck. Future datasets should therefore provide full input files consistent with their target DFT codes to enable modeling of all components required to initialize DFT calculations, including the charge density itself, augmentation occupancies, PAW charges, and SCF solver settings (see Appendix D for details). This would enable models to be evaluated in realistic and directly comparable end-to-end ML-accelerated DFT workflows. This is especially important for magnetic materials, for which general ML density models have not been developed and utilized. Extending ML-initialized DFT to this regime will require spin-polarized benchmarks with reproducible input files and models that predict spin-resolved charge densities rather than only the total valence density.

**Practical Utility of ML Density Models.** Our results demonstrate the importance of model efficiency and inference speed for computational and energy savings. We have addressed this by aligning ELECTRAFI with the canonical plane-wave basis set definition for periodic DFT. Rather than learning global reciprocal space structure with additional neural networks, ELECTRAFI uses closed-form Fourier structure to delegate global behavior to analytic transforms. The scalability of this approach has clear analogues in other domains, such as magnetic resonance imaging (MRI), where images are reconstructed from k-space measurements via large-scale inverse FFTs (Fessler, 2010; Griswold et al., 2002; Lustig et al., 2007; Knopp et al., 2007). Accordingly, even as end-to-end DFT acceleration benchmarks mature, ELECTRAFI's efficiency is immediately useful for downstream settings using charge density as a descriptor, such as molecular dynamics (de Blasio et al., 2023) where densities are constructed repeatedly, or in analyses of reactions and chemical bonding based on electronic structure (Koch et al., 2021). Generally, acceleration of electronic structure workflows is an extremely valuable use case, and ELECTRAFI advances the Pareto front of accuracy versus efficiency for scalable charge density prediction in periodic materials. But our results also point to a broader lesson for the field: grid-level accuracy or SCF step counts alone are insufficient predictors of real computational savings, and density models should be co-designed and integrated with DFT codes from the outset, explicitly accounting for inference cost and SCF behavior if acceleration is to be realized in practice.

## Acknowledgements

The authors acknowledge financial support from the Independent Research Foundation Denmark with grant no. 3164-00297B (ADANA), 2035-00232B (TeraBatt) and 0217-00326B (DELIGHT); the Novo Nordisk Foundation with grant number NNF25OC0101622(AutoMLP) and NNF24OC0089800; and the Pioneer Center for Accelerating P2X Materials Discovery (CAPeX), DNRF grant number P3.

## Impact Statement

This work aims to advance machine learning methods for electronic structure calculations that improve the efficiency of such calculations in crystals and other periodic materials. Such methods may reduce computational cost and energy use in large-scale materials simulation, with potential applications in materials discovery for energy, catalysis, semiconductors, and related technologies.

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

## Software and Data

The full codebase for ELECTRAFI and its associated experiments is publicly available at: `https://github.com/Jotels/ELECTRAFI`, under the license specified in the repository.

The DFT calculations in this paper have been performed using the ab-initio total-energy and molecular-dynamics program VASP (Vienna ab-initio simulation program) developed at the Fakultät für Physik of the Universität Wien (Kresse & Furthmüller, 1996; Kresse & Joubert, 1999).

# A. Notation and Plane-Wave Background

## A.1. Notation

Table 3 summarizes the main notation used throughout the paper.

*Table 3.* Summary of notation used in the paper.

| Symbol | Meaning |
|---|---|
| $\mathbf{r} \in \mathbb{R}^3$ | Real-space position |
| $\rho(\mathbf{r})$ | Electron/charge density at position $\mathbf{r}$ |
| $\rho_{\text{pred}}, \rho_{\text{ref}}$ | Predicted and reference charge densities |
| $\Omega \subset \mathbb{R}^3$ | Periodic simulation cell |
| $|\Omega|$ | Volume of the simulation cell |
| $\Delta v$ | Real-space grid voxel volume |
| $N_e$ | Number of valence electrons in the simulation cell |
| $N_{\mathcal{N}}$ | Total number of Gaussian components in a structure |
| $\mathbf{a}_1, \mathbf{a}_2, \mathbf{a}_3$ | Direct lattice vectors |
| $\Lambda$ | Direct lattice generated by $\mathbf{a}_1, \mathbf{a}_2, \mathbf{a}_3$ |
| $\mathbf{R} \in \Lambda$ | Direct-lattice translation vector |
| $\Lambda^*$ | Reciprocal lattice |
| $\mathbf{G} \in \Lambda^*$ | Reciprocal-lattice / plane-wave vector |
| $\hat{\rho}(\mathbf{G})$ | Fourier coefficient of the density at $\mathbf{G}$ |
| $j$ | Gaussian component index |
| $w^{(j)}$ | Signed weight of Gaussian component $j$ |
| $\boldsymbol{\mu}^{(j)}$ | Center of Gaussian component $j$ |
| $\boldsymbol{\Sigma}^{(j)}$ | Covariance matrix of Gaussian component $j$ |
| $\mathbf{d}^{(j)}$ | Displacement of Gaussian $j$ from its host atom |
| $M$ | Number of Gaussian components per valence electron |
| $v_a$ | Valence electron count assigned to atom $a$ |
| $n_a = M v_a$ | Number of Gaussian components assigned to atom $a$ |
| $N$ | Number of atoms in the structure |
| $C$ | Channel width of the ELECTRAFI readout |
| $S, V, T$ | Scalar, vector, and tensor channels output by the backbone/readout |
| $\mathcal{L}$ | Training loss, here NMAE |

**A.2. Plane-wave representations in electronic structure theory**

In electronic structure calculations for periodic systems, plane waves are functions of the form

$$\phi_{\mathbf{G}}(\mathbf{r}) = \exp(i\mathbf{G} \cdot \mathbf{r}), \tag{17}$$

where $\mathbf{G}$ is a reciprocal-lattice vector. Plane waves form a complete, orthonormal basis for square-integrable functions defined over a periodic unit cell, making them a natural representation for periodic quantities such as the electronic charge density $\rho(\mathbf{r})$.

The theoretical justification for the plane-wave representation in crystalline systems follows from Bloch's theorem, which states that eigenstates of a periodic Hamiltonian can be written as a plane wave multiplied by a lattice-periodic function (Bloch, 1929). As a consequence, any lattice-periodic scalar field, including the valence charge density, admits a Fourier-series expansion

$$\rho(\mathbf{r}) = \sum_{\mathbf{G}} \hat{\rho}(\mathbf{G}) \exp(i\mathbf{G} \cdot \mathbf{r}), \tag{18}$$

where $\hat{\rho}(\mathbf{G})$ are the plane-wave coefficients.

Plane waves are the dominant basis choice in periodic density functional theory (DFT) due to several favorable properties (Payne et al., 1992; Martin, 2020):

- **Systematic convergence**: Accuracy is controlled by a single kinetic-energy cutoff $E_{\text{cut}}$, yielding a variationally improvable basis.

- **Position independence**: The basis does not depend on atomic positions, eliminating basis-set superposition errors.

- **Efficient transforms**: Real- and reciprocal-space representations are connected via fast Fourier transforms (FFTs) with $\mathcal{O}(N \log N)$ scaling.

In plane-wave DFT implementations, the valence electron density is represented internally by its plane-wave coefficients, even when written to disk as a real-space grid (e.g. CHGCAR files). Core operations in the self-consistent field (SCF) cycle, such as evaluation of the kinetic energy, solution of Poisson's equation, and density mixing, are performed in reciprocal space. In projector-augmented-wave (PAW) methods, the smooth valence density is treated on the plane-wave grid, while atom-centered augmentation contributions are handled separately.

In SCF calculations, both the input and output charge densities are fundamentally reciprocal-space objects. Although densities are often visualized or stored on real-space grids, each SCF iteration repeatedly transforms between $\rho(\mathbf{r})$ and $\rho_{\mathbf{G}}$. Consequently, the effectiveness of an initial density guess is determined by how accurately it reproduces the correct global Fourier content, rather than only local real-space features.

This observation underlies recent machine-learning approaches that aim to accelerate SCF convergence by predicting charge densities directly (Jørgensen & Bhowmik, 2022; Koker et al., 2024). Most existing models, however, operate primarily in real space without explicit control over the structure of $\rho_{\mathbf{G}}$.

Plane waves are also used in condensed-matter and materials physics to represent Kohn-Sham orbitals, electrostatic potentials, phonon modes, and linear-response quantities. Their ubiquity stems from the fact that plane waves form the mathematically natural basis for translationally invariant systems.

## B. Model details

### B.1. Dyadic neighbor aggregation layers.

To form our Gaussian output parameters, we design a dyadic neighbor aggregation layer. This layer constructs scalar, vector and tensor output which we aggregate and use as the basis for $(w^{(j)}, \boldsymbol{\mu}^{(j)}, \boldsymbol{\Sigma}^{(j)})$.

Let $\mathcal{N}(i)$ be the neighbors of atom $i$, and let $\mathbf{n}_{i,e} \in \mathbb{R}^3$ denote the (PBC-aware) unit edge direction for $e \in \mathcal{N}(i)$. Each node $i$ carries scalar features $h_i \in \mathbb{R}^H$ and each incident edge $e$ carries scalar features $g_{i,e} \in \mathbb{R}^H$. A single dyadic layer produces, for each node $i$ and channel $c \in \{1, \ldots, C\}$,

$$S_{i,c} \in \mathbb{R}, \qquad \mathbf{v}_{i,c} \in \mathbb{R}^3, \qquad \mathbf{T}_{i,c} \in \mathbb{R}^{3 \times 3} \text{ (symmetric)}.$$

**Scalar branch.** We read out scalars channel-wise from node features:

$$S_{i,c} = g_c(h_i), \tag{19}$$

where $g_c$ is a MLP.

**Vector branch.** To enrich directional support before attention, we add learned directions to the geometric edge directions. For each edge $(i, e)$, we predict a free vector $\tilde{\mathbf{d}}_{i,e}^{(v)} \in \mathbb{R}^3$ and a scalar gate $a_{i,e}^{(v)} \in \mathbb{R}$ from $g_{i,e}$, and blend additively with the unit edge directions $\mathbf{n}_{i,e}$

$$\mathbf{n}_{i,e}^{(v)} = (1 - a_{i,e}^{(v)}) \mathbf{n}_{i,e} + a_{i,e}^{(v)} \tilde{\mathbf{d}}_{i,e}^{(v)}. \tag{20}$$

We then form edge-wise unit carrier vectors by normalizing $\mathbf{n}_{i,e}^{(v)}$:

$$\widehat{\mathbf{n}}_{i,e}^{(v)} = \frac{\mathbf{n}_{i,e}^{(v)}}{\|\mathbf{n}_{i,e}^{(v)}\|}. \tag{21}$$

For each channel $c$, we then compute neighbor weights $\alpha_{i,c,e}$ from $g_{i,e}$ with a softmax over $e \in \mathcal{N}(i)$ (so $\sum_e \alpha_{i,c,e} = 1$), and we aggregate

$$\mathbf{v}_{i,c} = \sum_{e \in \mathcal{N}(i)} \alpha_{i,c,e} \widehat{\mathbf{n}}_{i,e}^{(v)}$$
$$\mathbf{v}_{i,c} \leftarrow \frac{\mathbf{v}_{i,c}}{\|\mathbf{v}_{i,c}\|} \cdot m_{i,c}, \tag{22}$$

where $m_{i,c} = \phi_m(h_i) \in \mathbb{R}_{>0}$ is a scalar controlling the magnitude of the vector output of atom $i$ and channel $c$, produced by an MLP, $\phi_m$.

The additive term in (20) expands the directional span from the unit-edge cone to the two-dimensional subspace $\text{span}\{\mathbf{n}_{i,e}, \tilde{\mathbf{d}}_{i,e}^{(v)}\}$, enabling off-axis and lower-symmetry orientations that better capture charge density patterns not aligned with edges. This design is inspired by Qu & Krishnapriyan (2024), where force output is formed using learned edgewise vectors $\mathbf{d}_{i,e}$ predicted from edge features $g_{i,e}$. These are gated multiplicatively by the geometric direction through the Hadamard product $\mathbf{d}_{i,e} \odot \mathbf{n}_{i,e}$ before aggregation. Their gating preserves componentwise alignment with $\mathbf{n}_{i,e}$, and our additive carriers complement this by providing a richer, data-driven directional basis while retaining the geometric limit when $a_{i,e}^{(v)} = 0$ (pure $\mathbf{n}_{i,e}$). In practice, the attention in (22) then learns to combine these carriers into stable, geometry-consistent channel-wise directions.

**Tensor branch.** To construct the tensor output, we use a carrier vector construction that is analogous to Equation (20), but with separate gate $a_{i,e}^{(t)}$ and free vector $\tilde{\mathbf{d}}_{i,e}^{(t)}$. This defines unit direction carrier vectors $\widehat{\mathbf{n}}_{i,e}^{(t)}$, which we use to construct dyads

$$\mathbf{Q}_{i,e} = \widehat{\mathbf{n}}_{i,e}^{(t)} \widehat{\mathbf{n}}_{i,e}^{(t)\top} - \tfrac{1}{3} \mathbf{I}, \tag{23}$$

where tracelessness, i.e. $\text{tr}\, \mathbf{Q}_{i,e} = 0$, has been enforced by subtracting the unit-normalized $3 \times 3$ identity matrix $\mathbf{I}$. For each channel $c$, we compute neighbor weights $\beta_{i,c,e}$ (softmax over $e$, $\sum_e \beta_{i,c,e} = 1$) and combine the anisotropic attention-weighted tensor aggregations with an isotropic contribution:

$$\mathbf{T}_{i,c} = \sum_{e \in \mathcal{N}(i)} \beta_{i,c,e} \mathbf{Q}_{i,e} + \kappa_{i,c} \mathbf{I}, \tag{24}$$

where the isotropic contribution $\kappa_{i,c}\mathbf{I}$ controls the trace of $\mathbf{T}_{i,c}$ through a scalar predicted via an MLP $\tilde{g}$ on the node features, i.e. $\kappa_{i,c} = \tilde{g}(h_{i,c}) \in \mathbb{R}$.

## B.2. Backbone updates and cross-layer aggregation.

Between dyadic layers, we update node and edge features $(h_i, g_{i,e})$ by attention over the neighborhood $\mathcal{N}(i)$ and position-wise feed-forward transformations (FFN), mixing information across neighbors. Each dyadic layer $\ell = 0, 1, \ldots, L$ (including the input layer $\ell = 0$) produces its own triplet

$$S_{i,c}^{[\ell]}, \quad \mathbf{v}_{i,c}^{[\ell]}, \quad \mathbf{T}_{i,c}^{[\ell]}.$$

We aggregate along the layer axis using channel-wise, per-node softmax weights obtained from the layer readouts:

$$\alpha_{i,c}^{[\ell]} = \mathrm{softmax}\left(A_{i,c}^{[\ell]}\right) = \frac{\exp\left(A_{i,c}^{[\ell]}\right)}{\sum_{m=0}^{L}\exp\left(A_{i,c}^{[m]}\right)}. \tag{25}$$

Where $A_{i,c}^{[\ell]}$ are logits obtained from a linear layer applied to the node readout $\mathbf{r}_i^{[\ell]}$ at layer $\ell$ and node $i$, i.e. $A_{i,c}^{[\ell]} = \left(W^{\top}\mathbf{r}_i^{[\ell]} + \mathbf{b}\right)_c$, The final scalar, vector, and tensor channels are convex combinations over layers:

$$S_{i,c} = \sum_{\ell=0}^{L}\alpha_{i,c}^{[\ell]}S_{i,c}^{[\ell]},$$

$$\mathbf{v}_{i,c} = \sum_{\ell=0}^{L}\alpha_{i,c}^{[\ell]}\mathbf{v}_{i,c}^{[\ell]}, \tag{26}$$

$$\mathbf{T}_{i,c} = \sum_{\ell=0}^{L}\alpha_{i,c}^{[\ell]}\mathbf{T}_{i,c}^{[\ell]}.$$

With the global mapping $j \equiv (i, c)$, we then set

$$\mathbf{d}^{(j)} \leftarrow \mathbf{v}_{i,c}, \quad \mathbf{A}^{(j)} \leftarrow \mathbf{T}_{i,c}, \quad S^{(j)} \leftarrow f_w\left(S_{i,c}\right). \tag{27}$$

These feed into Eqs. (13), (14), and (15) to construct $\left(\mu^{(j)}, \Sigma^{(j)}, w^{(j)}\right)$ for all Gaussians indexed by $j = 1, \ldots, N_{\mathcal{N}}$, which we then use in Eq. (9) and the subsequent IFFT to realize $\rho(\mathbf{r})$.

## B.3. Multi-valence atomic representations.

For faithful comparison to ab initio reference densities, we adopt the same per-element valence conventions as those used to generate the datasets (e.g., PAW/USPP pseudopotentials in VASP that provide alternative ZVAL options such as $W{:}6$ or $W{:}14$). For each element $Z$ we therefore use a small admissible set $\{v^{(0)}(Z), v^{(1)}(Z), \ldots\}$ and use the element-specific selection prescribed by the underlying pseudopotential setup. The chosen per-atom valence $v_a$ is therefore directly derived from default VASP input files, and directly sets the Gaussian allotment $n_a = M\,v_a$, so that the total component count $\sum_a n_a$ is consistent with the reference valence specification. To reflect these physical conventions in the learned representation, element-valence pairs are endowed with distinct embeddings. This enables ELECTRAFI to capture density patterns appropriate to each valence configuration (e.g., semicore-including vs. semicore-excluding) through the embedding as well as through the valence-sensitive allocation of Equation (12).

## C. Rotation Augmentation and Equivariance

ELECTRAFI uses a modified EScAIP backbone that is not explicitly constrained to be rotation-equivariant. Consequently, exact transformation rules for the predicted Gaussian parameters are not guaranteed by construction: under a global rotation of the input coordinates, the predicted displacements and covariance matrices are not forced architecturally to transform as

$$d^{(j)} \mapsto Q d^{(j)}, \tag{28}$$

$$\Sigma^{(j)} \mapsto Q \Sigma^{(j)} Q^{\top}, \tag{29}$$

for a rotation matrix $Q \in \mathrm{SO}(3)$.

In the DFT initialization setting studied in this work, this limitation is mitigated by the fact that routine plane-wave DFT workflows provide crystal structures in a fixed lattice frame. The model is therefore trained and evaluated in the same coordinate convention used by the target DFT code, and exact equivariance is not required for the reported end-to-end acceleration experiments. Moreover, when ELECTRAFI is used as an SCF initializer, any residual error in the predicted density is corrected by the subsequent self-consistent DFT procedure. Such errors may affect convergence speed, but not the final converged DFT solution.

Nevertheless, equivariance and rotation robustness are important when charge density models are used outside fixed-frame DFT initialization workflows, for example as descriptors for downstream analysis or as intermediate representations in other learning systems. We therefore performed an additional rotation-augmentation experiment on the ECD-HSE06 benchmark to test whether ELECTRAFI can learn stable behavior under arbitrary rotations.

We compare three settings. The baseline model is trained and evaluated on the original, unrotated structures. In the second setting, the model is trained on unrotated structures but evaluated on randomly rotated test structures. In the third setting, random rotations are applied during both training and evaluation. The results are shown in Table 4.

*Table 4.* Effect of random rotation augmentation on ELECTRAFI using the ECD-HSE06 benchmark. The baseline is trained and evaluated in the original lattice frame. Evaluating an unaugmented model on rotated structures increases the test error, while training with rotations recovers and slightly improves the rotated-test performance.

| Setting | Train data | Test data | Train NMAE [%] | Test NMAE [%] |
|---|---|---|---|---|
| Baseline | Unrotated | Unrotated | 0.70 | 1.35 |
| Experiment 1 | Unrotated | Rotated | 0.70 | 1.69 |
| Experiment 2 | Rotated | Rotated | 0.88 | 1.33 |

When the model is trained only on the original lattice-frame structures and tested on randomly rotated structures, the test error increases from $1.35\%$ to $1.69\%$ NMAE. This is expected because the architecture is not exactly equivariant and has not been exposed to arbitrary coordinate-frame rotations during training. However, when random rotations are included during training, the rotated-test error decreases to $1.33\%$ NMAE, slightly improving over the unrotated baseline. This indicates that the attention-based Cartesian backbone can learn rotation-robust density prediction from data augmentation, consistent with prior observations that unconstrained attention-based atomistic models can recover effectively equivariant behavior when trained with sufficiently rich geometric information and rotational augmentation (Qu & Krishnapriyan, 2024; Qu et al., 2026; Mazitov et al., 2025).

## D. Experiments, DFT setup and CHGCAR handling

**Data.**    The main ELECTRAFI model is trained on the full Materials Project (MP) distribution (MP-Full), which we split into 117,876 training structures, 512 validation structures and 2,000 test structures. This is similar to the main periodic model published by Koker et al. (2024). This version of ELECTRAFI is also evaluated on 2000 structures from GNoME, which probes generalization to a broader distribution. We note that for both the MP-Full and GNoME datasets, we use different structures than those used in Koker et al. (2024). Instead, the GNoME files are identical to those used in Chen et al. (2025), while we use a different split of the Materials Project database due to reproducibility issues with the Materials Project IDs. To ensure fair comparison, we thus perform our own evaluations and use the ChargE3Net model trained on the full Materials Project database, as published in Koker et al. (2024). However, we note that the results are essentially identical to the published ChargE3Net performance in both Koker et al. (2024) and Chen et al. (2025), and the NMAE[%] differs only on the second decimal in both.

**Non-magnetic filter.**    For the SCF reduction experiments in section 4.1, we follow Koker et al. (2024) and restrict reinitialized DFT experiments to non-magnetic structures. Specifically, we retain only test set entries with an overall absolute magnetic moment below $0.1\,\mu_B$ and absolute site-resolved magnetic moments below $0.1\,\mu_B$ on every atom. Applying this filter yields 980 structures from the MP-Full test set and 1314 structures from the GNoME benchmark.

**Reconstructing MP-Full inputs from task documents.**    For MP-Full we start from the same Materials Project charge density entries used in Koker et al. (2024). For each material, we call the Materials Project API to retrieve the charge density document together with the underlying VASP TaskDoc that originally generated the CHGCAR. From this TaskDoc we reconstruct:

- **POSCAR**: we write the structure stored in the task input directly to VASP `POSCAR`.

- **INCAR**: we take the recorded input parameters and write them verbatim to `INCAR`.

- **KPOINTS**: if explicit $k$-point data are present, we reconstruct the original `KPOINTS`; otherwise we let VASP infer $k$-points from the INCAR settings (e.g. KSPACING).

- **POTCAR**: we use the `potcar_spec` field to recover the original PAW symbols and build the `POTCAR` files specifying the pseudopotentials, using a local directory containing the legacy PBE PAW potentials used in the original Materials Project workflow.

All MP-Full recomputations are run with VASP 5.4.4. For the SCF experiments we minimally edit the regenerated INCAR: we remove parallelization hints such as NPAR, NCORE, KPAR, and NSIM, enforce `ISTART = 0` to always start from scratch, set `LCHARG = .TRUE.` to write CHGCAR, and vary only `ICHARG` to distinguish SAD (`ICHARG = 2`), true-density-init, and ML-init runs (`ICHARG = 1`). We do not override the FFT grid parameters (NGXF/NGYF/NGZF) in INCAR; instead, we parse these directly from the reference CHGCAR files.

**GNoME setup via `MPStaticSet`.**    For the GNoME benchmark, the public dataset from Chen et al. (2025) provides CHGCAR files. However, these were computed using newer pseudopotentials than the ones used for the legacy MP dataset. Thus, for more straightforward comparison and interpretable results, we calculate new charge density files that are consistent with those used in MP. To do this, we parse the atomic structure from each CHGCAR using ASE (Larsen et al., 2017), and then use `pymatgen`'s `MPStaticSet` to generate MP-style static calculations from this structure. This automatically produces `INCAR`, `KPOINTS`, and `POTCAR` (provided a local copy of the proper legacy pseudopotentials is available) consistent with the Materials Project defaults: PBE exchange–correlation, a 520 eV plane-wave cutoff, Monkhorst–Pack meshes at approximately the same $k$-point density used by the Materials Project (of order $10^2$ points $\text{Å}^{-3}$), and the same PAW pseudopotential library as above.

To match the DFT+$U$ treatment of Koker et al. (2024), we post-process the generated INCAR files and, whenever `LDAU = .TRUE.`, set

$$\text{LMAXMIX} = \begin{cases} 6, & \text{if any LDAUL channel uses } l = 3 \text{ (f-orbitals),} \\ 4, & \text{else if any LDAUL channel uses } l = 2 \text{ (d-orbitals)}, \end{cases}$$

leaving LMAXMIX unchanged otherwise. As noted in Koker et al. (2024), this is theoretically incorrect but necessary for better comparison, since it reproduces the MP convention of using LMAXMIX= 4 for $d$-elements and LMAXMIX= 6 for $f$-elements only when DFT+$U$ is active. All other DFT+$U$ parameters (Hubbard $U$ values, projectors) are those supplied by `MPStaticSet`, which follow the Materials Project configuration.

**CHGCAR initialization and PAW augmentation.** For both MP-Full and GNoME, we distinguish three types of SCF runs:

- **Default/SAD runs** (`ICHARG = 2`), which produce a converged reference CHGCAR starting from the standard superposition-of-atomic-densities (SAD) initialization.

- **True-density-init runs** (`ICHARG = 1`), which are initialized from the converged "ground-truth" CHGCAR of the corresponding default run.

- **ML-init runs** (`ICHARG = 1`), which are initialized from a CHGCAR whose valence density grid is replaced by a machine-learning prediction from either ChargE3Net or ELECTRAFI.

In VASP/PAW, CHGCAR includes both the smooth grid density and one-center PAW augmentation occupancies. Consistent with prior work (Jørgensen & Bhowmik, 2022; Koker et al., 2024; Elsborg et al., 2026), we replace only the smooth grid component with the ML prediction and keep augmentation occupancies from the corresponding reference workflow, thereby isolating the impact of the ML-predicted smooth density on SCF convergence. Since all three types of density initialization use the same reference augmentation occupancies for the MP, the comparison between SCF reductions due to better valence density guesses is straightforward. However, in a scenario where access to these augmentation occupancies cannot be assumed, it would be necessary to also model the augmentation occupancies directly. This issue exists for all real-space ML density models aimed at plane-wave codes (Kim & Ahn, 2024; Cheng & Peng, 2024; Koker et al., 2024; Fu et al., 2024; Elsborg et al., 2026). While CHGCAR stores the smooth real-space density on a grid, the augmentation occupancies correspond to atom-centered projector expansions whose dimensionality, angular-momentum resolution, and symmetry constraints are defined implicitly by the PAW datasets rather than by the charge file itself. As a consequence, the number of coefficients per atom, their association with specific $(l, m)$ channels, and the rotational transformation rules they must obey are not directly inferable from CHGCAR files. This further complicates any attempt to learn augmentation occupancies in a transferable manner. Recent work has explored modeling the augmentation occupancies for specific systems (Focassio et al., 2024), but no method currently exists that is fully general and that uses, e.g., a graph neural network backbone that can be trained to work across many different materials and structures. We aim to address this aspect shortly after the present work.

**Electron-count normalization.** ELECTRAFI enforces electron-count normalization by construction. To ensure a fair comparison, we apply the same post-processing to ChargE3Net by rescaling its predicted total charge density to match the integrated electron number implied by the pseudopotentials for the corresponding structure. This normalization yields slightly lower NMAE and slightly faster SCF convergence for ChargE3Net and ensures that both methods are evaluated under consistent electron-number constraints.

# E. Datasets and training settings

Table 5 shows the dataset size for each dataset evaluated in this work. Aside from the GNoME results, all results reported in Table 1 of the main text were obtained by training exclusively on the corresponding dataset and evaluating on its test set. For GNoME, results were obtained by evaluating an ELECTRAFI model trained on MP-Full. In Tables 6 and 7 we report the common settings used across all models trained, as well as the settings that were tuned for specific datasets, respectively. Most notably, the ECD-HSE06 model uses a larger hidden size and more Gaussians per electron, which we found to yield higher accuracies, presumably because the HSE06 densities are more complicated than PBE densities. Furthermore, MP-Mixed and MP-Full contain larger structures than the other datasets (Max nodes per batch is equivalent to the maximum number of atoms in a single structure per dataset). Thus, these also use a larger cutoff radius and larger maximum neighbors. The MP-Full dataset is significantly larger than any other dataset which is why the model for this data corpus was trained for longer.

*Table 5.* Dataset splits used in this work. *The GNoME test set was evaluated as an out-of-distribution dataset for the full ELECTRAFI model trained on MP-Full, and thus has no training or validation structures.

|  | MP-Full | GNoME | ECD-HSE06 | MP-Mixed | Cubic |
|---|---|---|---|---|---|
| **# Training structures** | 117,876 | — | 5,647 | 14,000 | 14,421 |
| **# Validation structures** | 512 | — | 500 | 1,050 | 1,000 |
| **# Test structures** | 2,000 | 2,000* | 1,000 | 1,050 | 1,000 |

*Table 6.* Model architecture, physical assumptions, and optimization settings shared across all data splits.

| Category | Setting |
|---|---|
| *EScAIP Backbone and Architecture* | |
| Number of attention layers | 2 |
| Attention heads | 32 |
| Distance cutoff function | Gaussian |
| Angle embeddings | True |
| Irrep aggregation | Softmax |
| Normalization | LayerNorm |
| Activation function | GELU |
| *Graph Construction and Symmetry* | |
| Periodic boundary conditions | Enabled |
| On-the-fly graph construction | Enabled |
| Maximum radius | Set by cutoff (split-specific, see Table 7) |
| Max neighbors | Split-specific (see Table 7) |
| Strict neighbor enforcement | Enabled |
| *Optimization* | |
| Optimizer | Muon (Jordan et al., 2024) (2D weight matrices) & Adam (everything else) (Kingma, 2014) |
| Initial learning rate | $3 \times 10^{-4}$ |
| Learning rate schedule | Exponential decay |
| Gradient clipping | Disabled |
| Precision | FP32 |

*Table 7.* Split-specific settings that differ across datasets. Only parameters that change between splits are shown.

| Setting | ECD-HSE06 | Cubic | MP-Mixed | MP-Full |
|---|---|---|---|---|
| Backbone hidden size (C) | 2160 | 2200 | 2160 | 2160 |
| Atom embedding size (C) | 2160 | 2200 | 2160 | 2160 |
| Gaussians per electron ($M$ - see Equation 12) | 120 | 100 | 120 | 120 |
| Cutoff radius (Å) | 20 | 20 | 30 | 30 |
| Max neighbors | 200 | 128 | 200 | 200 |
| Max nodes per batch | 20 | 64 | 154 | 154 |
| Learning rate decay $\gamma$ | 0.99 | 0.95 | 0.95 | 0.70 |
| Training epochs | 140 | 40 | 20 | 10 |

# F. Error trends across elements and datasets

### F.1. Element-wise performance analysis

To better understand where different modeling approaches succeed or fail, we analyze element-resolved errors in the predicted valence charge densities. This analysis aims to identify systematic performance trends across chemical species, rather than focusing solely on aggregate dataset-level metrics.

Figure 5 (next page) shows element-resolved valence-density errors for ChargE3Net and ELECTRAFI on the MP-Full and GNoME test sets. For each model and dataset, the average NMAE is computed over all structures containing a given element, without controlling for stoichiometric fraction. The bottom row reports the per-element error difference between the two models, highlighting systematic regimes where one model consistently outperforms the other across datasets.

For the GNoME data (Figure 5f), the error trends are roughly the same for ELECTRAFI and ChargE3Net, although ELECTRAFI generally has higher error on most elements. However, on MP-Full (Figure 5e), where both models perform significantly better, the element-wise performance trends are consistent with qualitative differences in the electronic structure regimes emphasized by the two models. ELECTRAFI outperforms ChargE3Net on alkali and alkaline-earth metals (Rb, Sr, Ba), halogens (Br, I), and several heavy elements (Au, Pt, Tl, Pa, Ac), which typically appear in ionic, closed-shell, or weakly directional bonding environments, where the valence charge density is comparatively smooth and dominated by low-to mid-spatial-frequency components. In these cases, accurately placing charge at the correct global length scales appears more important than resolving highly localized or anisotropic bonding features, favoring ELECTRAFI's inductive bias. In contrast, ChargE3Net performs better on light covalent elements (B, C) and open-shell transition metals (Fe, Co, Mn, Ru, Re, Ir, U), where valence densities exhibit stronger localization, directional bonding, and higher spatial-frequency structure associated with covalent bonds, crystal-field effects, and partially filled d or f shells. The particularly strong performance split for Ac suggests sensitivity to the degree of active valence-shell participation and dataset sparsity in heavy-element regimes. While these trends are reported without controlling for stoichiometry or chemical environment, they indicate that the two models emphasize complementary aspects of valence-density reconstruction, rather than one being uniformly superior across all electronic-structure regimes.

### F.2. Dataset distributional shift

We compare the atom-type distributions of MP-Full and GNoME in figure 6 to understand the origin of the chemistry-dependent generalization gap better. The comparison reveals a substantial distributional shift between the two datasets. MP-Full is dominated by oxide-containing compounds, which often exhibit relatively smooth and ionic charge densities, a regime that is particularly favorable for ELECTRAFI's Fourier-based representation. In contrast, GNoME spans a much broader and more uniform region of chemical space, containing a larger diversity of bonding environments and localized density features. This shift is consistent with the element-wise performance degradation observed on GNoME. ChargE3Net, which performs local message passing directly on probe points in real space, appears more robust to these changes in chemical composition and local bonding character, albeit at substantially higher computational cost.

### F.3. Spectral decomposition of density errors

To further characterize the generalization gap between MP-Full and GNoME, we analyze the radial power spectra of the ground-truth charge densities and model predictions in reciprocal space. Figure 7 reports the normalized mean spectral power as a function of $|G|$, together with the mean spectral deviation from the ground truth for ELECTRAFI and ChargE3Net. Compared to MP-Full, GNoME exhibits reduced relative power in the lowest-$|G|$ modes and a larger contribution from intermediate- and high-$|G|$ components. In real space, these modes correspond to shorter length scales, with $|G| \approx 2.5\ \text{Å}^{-1}$ corresponding to $\lambda = 2\pi/|G| \approx 2.5\ \text{Å}$, i.e., a characteristic local coordination scale.

This spectral shift is consistent with both the chemical and structural differences between the datasets. GNoME contains a broader distribution of elements and bonding environments, and its structures are substantially smaller than those in MP-Full, with mean/max numbers of atoms of approximately $9/56$ for GNoME compared to $24/154$ for MP-Full. Smaller cells reduce the relative contribution of extended, smooth density variations and increase the importance of localized coordination-scale features. The model spectra further support the interpretation that ELECTRAFI and ChargE3Net operate in complementary regimes: ELECTRAFI closely tracks the dominant low-frequency structure but shows larger deviations in localized, lower-$|G|$ components, whereas ChargE3Net is comparatively more robust in these local regimes due to its probe-based real-space representation.

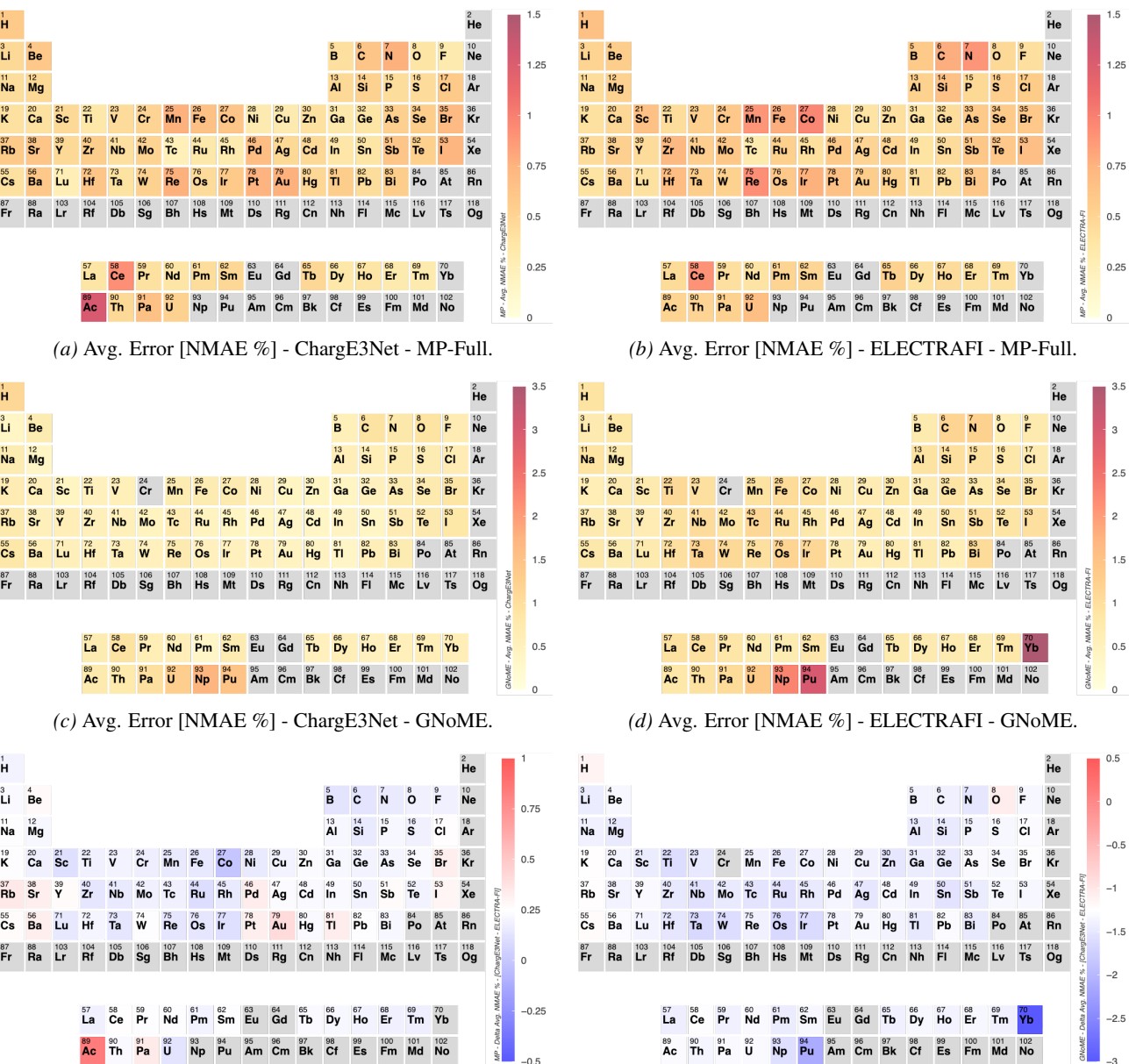

*(a)* Avg. Error [NMAE %] - ChargE3Net - MP-Full.

*(b)* Avg. Error [NMAE %] - ELECTRAFI - MP-Full.

*(c)* Avg. Error [NMAE %] - ChargE3Net - GNoME.

*(d)* Avg. Error [NMAE %] - ELECTRAFI - GNoME.

*(e)* Delta between Avg. Error [NMAE %] for ChargE3Net vs ELECTRAFI across elements in MP-Full. Red indicates higher error for ChargE3Net, blue indicates higher error for ELECTRAFI.

*(f)* Delta between Avg. Error [NMAE %] for ChargE3Net vs ELECTRAFI across elements in GNoME. Red indicates higher error for ChargE3Net, blue indicates higher error for ELECTRAFI.

*Figure 5.* Error trends across elements for ELECTRAFI and ChargE3Net on the MP-Full and GNoME test sets. The error for each element has been calculated as the average error for all structures including the element.

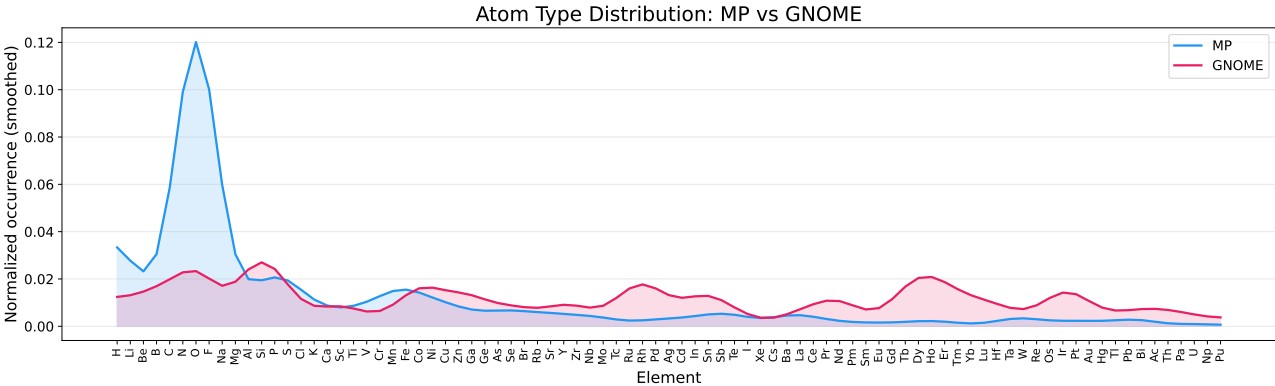

*Figure 6.* Normalized atom-type distributions for MP-Full and GNoME. MP-Full is dominated by oxide-containing compounds, whereas GNoME exhibits a substantially broader and more uniform distribution across chemical space.

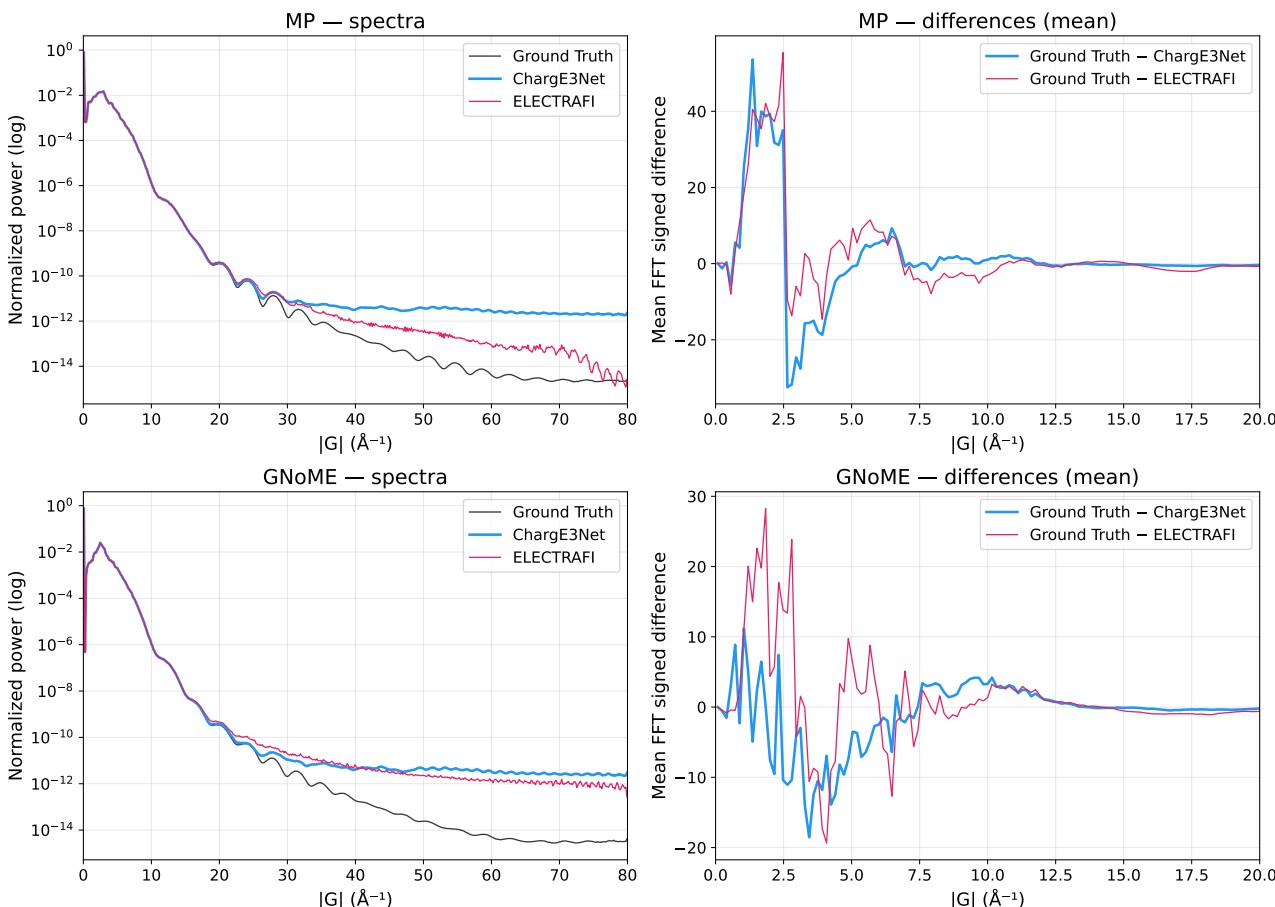

*Figure 7.* Radial reciprocal-space power spectra of charge densities for MP-Full and GNoME. The spectra are computed by averaging the Fourier-space power over shells of constant $|G|$. **Top Left:** Normalized mean radial power spectra for MP-Full, comparing the ground-truth densities against ELECTRAFI and ChargE3Net predictions. Both models reproduce the dominant low-$|G|$ structure, while deviations become more pronounced at larger $|G|$. **Top Right:** Mean signed spectral difference between the predicted and reference spectra for MP-Full. Both models struggle to capture the details in the low $|G|$ region, ChargE3Net more closely matching intermediate-$|G|$ components. ELECTRAFI has similar or better accuracy than CHargE3Net but varies more. **Bottom Left:** Normalized mean radial power spectra for GNoME. Relative to MP-Full, the spectra exhibit comparatively stronger intermediate- and high-$|G|$ contributions, consistent with more localized coordination-scale density structure. **Bottom Right:** Mean signed spectral difference for GNoME. ELECTRAFI shows larger deviations in intermediate-$|G|$ regions, while ChargE3Net remains comparatively more stable across localized spectral modes, supporting the interpretation that the two models possess complementary inductive biases.

# G. Density prediction examples

## G.1. Best from MP-Full test set - AlMg$_{30}$NaO$_{32}$

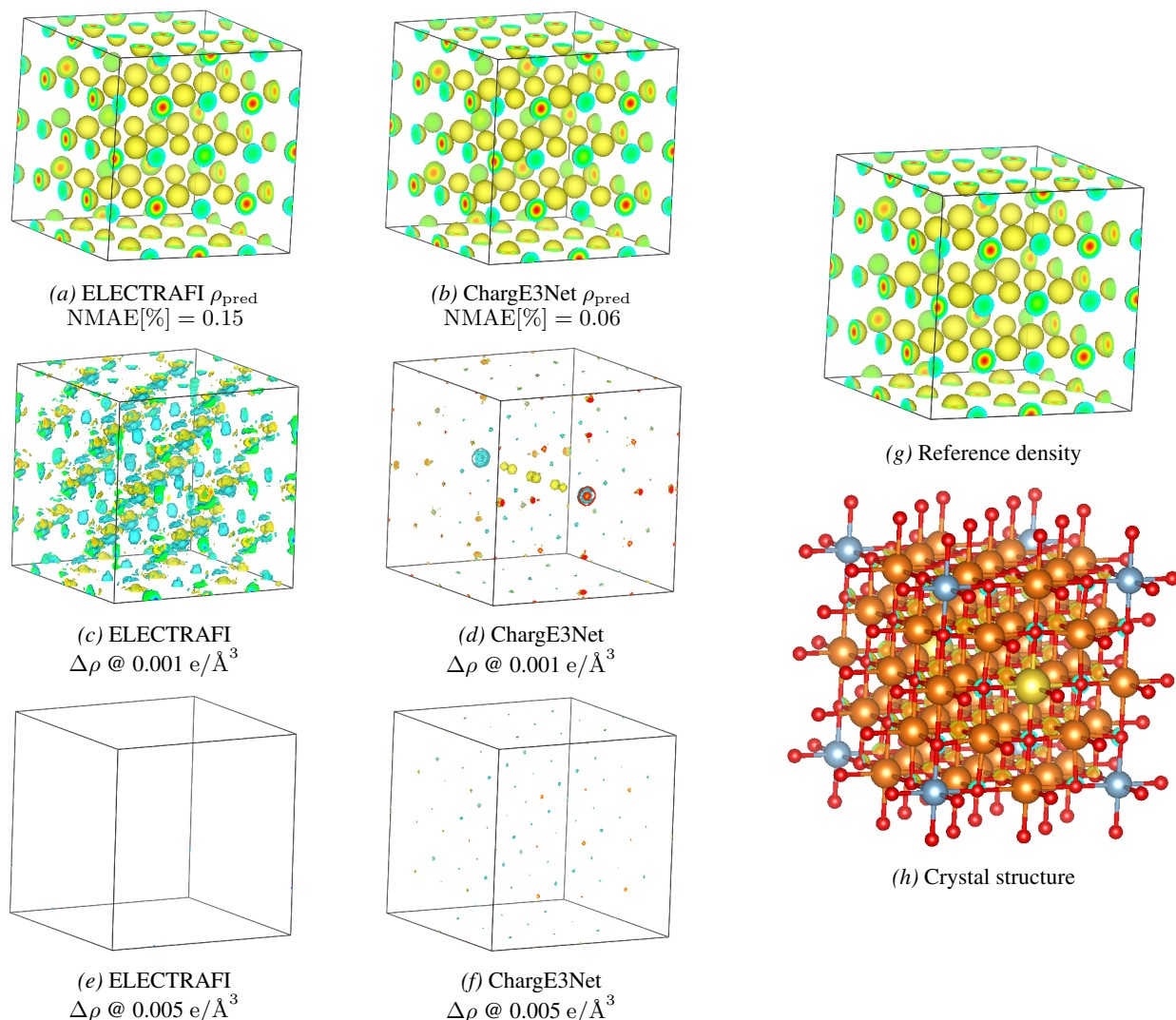

*(a)* ELECTRAFI $\rho_{\mathrm{pred}}$
NMAE[%] $= 0.15$

*(b)* ChargE3Net $\rho_{\mathrm{pred}}$
NMAE[%] $= 0.06$

*(g)* Reference density

*(c)* ELECTRAFI
$\Delta\rho$ @ 0.001 e/Å$^3$

*(d)* ChargE3Net
$\Delta\rho$ @ 0.001 e/Å$^3$

*(e)* ELECTRAFI
$\Delta\rho$ @ 0.005 e/Å$^3$

*(f)* ChargE3Net
$\Delta\rho$ @ 0.005 e/Å$^3$

*(h)* Crystal structure

*Figure 8.* **Lowest error structure from the MP-Full test set.** (a,b) Predicted densities for ELECTRAFI and ChargE3Net. (c–f) $\Delta\rho = \rho_{\mathrm{pred}} - \rho_{\mathrm{ref}}$ at two iso-levels. (g) Reference density from SCF calculations. (h) Crystal structure AlMg$_{30}$NaO$_{32}$.

### G.2. Worst from MP-Full test set - NH$_4$

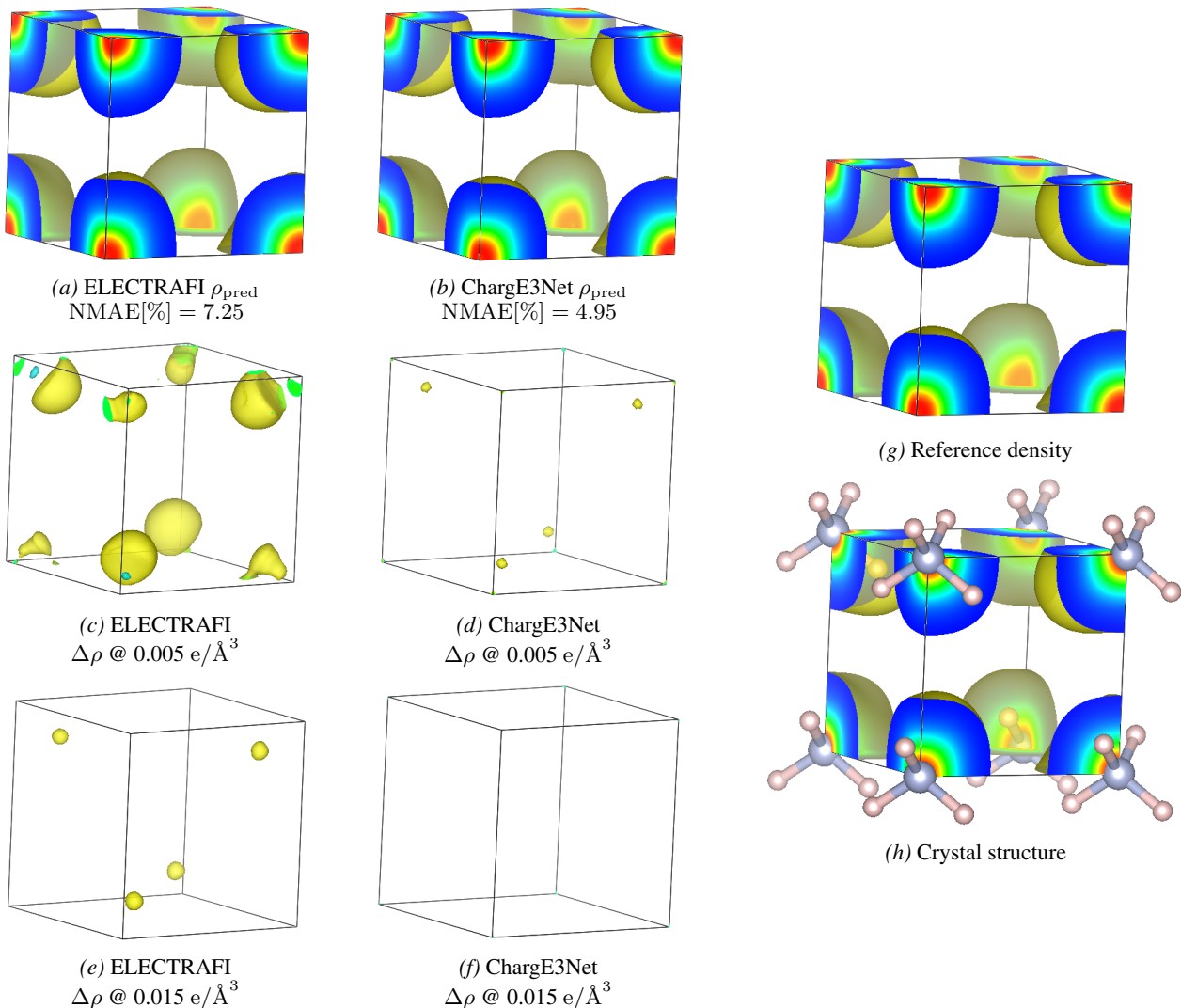

*(a)* ELECTRAFI $\rho_{\text{pred}}$
NMAE[%] = 7.25

*(b)* ChargE3Net $\rho_{\text{pred}}$
NMAE[%] = 4.95

*(c)* ELECTRAFI
$\Delta\rho$ @ 0.005 e/Å$^3$

*(d)* ChargE3Net
$\Delta\rho$ @ 0.005 e/Å$^3$

*(e)* ELECTRAFI
$\Delta\rho$ @ 0.015 e/Å$^3$

*(f)* ChargE3Net
$\Delta\rho$ @ 0.015 e/Å$^3$

*(g)* Reference density

*(h)* Crystal structure

*Figure 9.* **Highest error structure from the MP-Full test set.** (a,b) Predicted densities for ELECTRAFI and ChargE3Net. (c–f) $\Delta\rho = \rho_{\text{pred}} - \rho_{\text{ref}}$ at two iso-levels. (g) Reference density from SCF calculations. (h) Crystal structure for NH$_4$.

### G.3. Best from GNoME test set - $F_{28}Hf_4Tm_3Y$

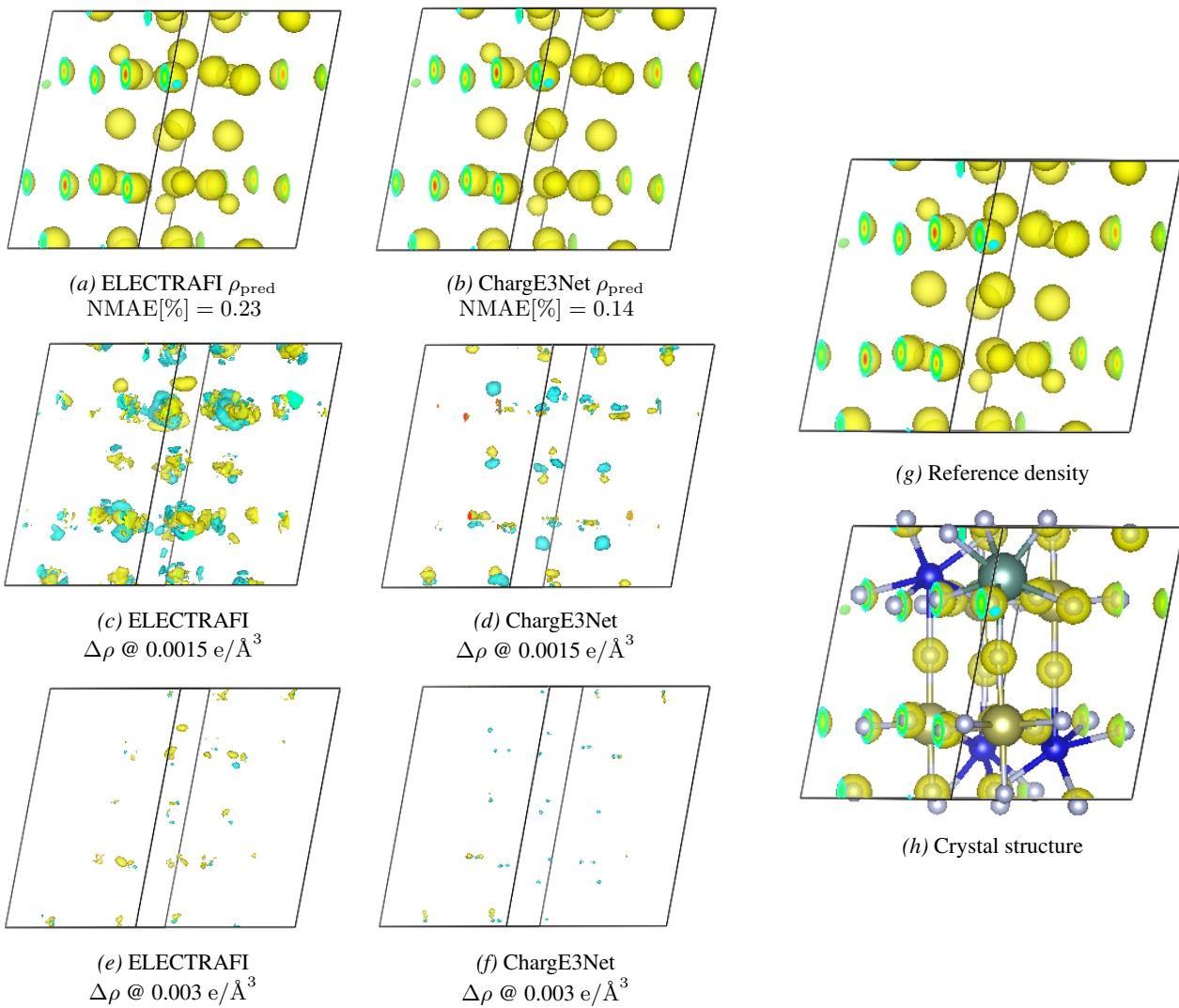

*(a)* ELECTRAFI $\rho_{\text{pred}}$
NMAE$[\%] = 0.23$

*(b)* ChargE3Net $\rho_{\text{pred}}$
NMAE$[\%] = 0.14$

*(c)* ELECTRAFI
$\Delta\rho$ @ $0.0015$ e/Å$^3$

*(d)* ChargE3Net
$\Delta\rho$ @ $0.0015$ e/Å$^3$

*(e)* ELECTRAFI
$\Delta\rho$ @ $0.003$ e/Å$^3$

*(f)* ChargE3Net
$\Delta\rho$ @ $0.003$ e/Å$^3$

*(g)* Reference density

*(h)* Crystal structure

*Figure 10.* **Lowest error structure from the GNoME test set.** (a,b) Predicted densities for ELECTRAFI and ChargE3Net. (c–f) $\Delta\rho = \rho_{\text{pred}} - \rho_{\text{ref}}$ at two iso-levels. (g) Reference density from SCF calculations. (h) Crystal structure for $F_{28}Hf_4Tm_3Y$.

### G.4. Worst from GNoME test set - CeErS$_2$

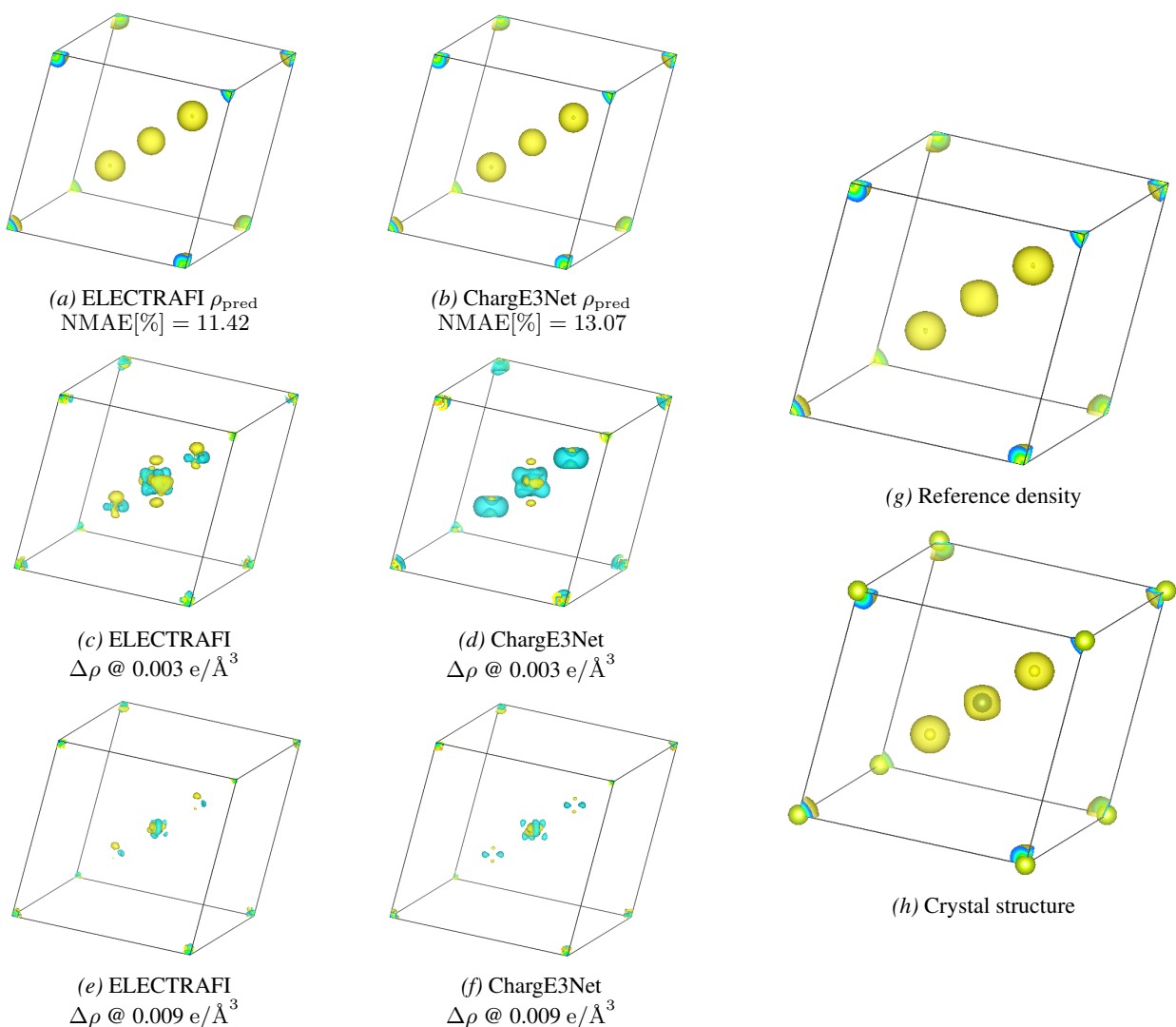

*(a)* ELECTRAFI $\rho_{\text{pred}}$
NMAE[%] = 11.42

*(b)* ChargE3Net $\rho_{\text{pred}}$
NMAE[%] = 13.07

*(c)* ELECTRAFI
$\Delta\rho$ @ 0.003 e/Å$^3$

*(d)* ChargE3Net
$\Delta\rho$ @ 0.003 e/Å$^3$

*(e)* ELECTRAFI
$\Delta\rho$ @ 0.009 e/Å$^3$

*(f)* ChargE3Net
$\Delta\rho$ @ 0.009 e/Å$^3$

*(g)* Reference density

*(h)* Crystal structure

*Figure 11.* **Highest error structure from the GNoME test set.** (a,b) Predicted densities for ELECTRAFI and ChargE3Net. (c–f) $\Delta\rho = \rho_{\text{pred}} - \rho_{\text{ref}}$ at two iso-levels. (g) Reference density from SCF calculations. (h) Crystal structure for CeErS$_2$.

