# OpenReview forum: "Global Plane Waves from Local Gaussians: Periodic Charge Densities in a Blink"
_ICML.cc/2026/Conference — ICML 2026 regular_

### Official Review · Reviewer_degT · 2026-03-08

**Soundness:** 3
**Presentation:** 2
**Significance:** 3
**Originality:** 3
**Overall Recommendation:** 5
**Confidence:** 3

**Summary:**

This paper introduces ELECTRAFI, a machine learning model for predicting periodic electron densities in crystalline materials. The model represents the density as a mixture of anisotropic Gaussians in real space. The paper demonstrates that the floating gaussian representation achieves state-of-the-art accuracy on periodic benchmarks, while achieving great speedup compared with grid-based density model.

**Compliance With Llm Reviewing Policy:**

Affirmed.

**Key Questions For Authors:**

1. For Weakness 1, the author could provide a comparison with a model directly estimating the Fourior coefficients.

2. For Weakness 2, a notation table, and a background introduction, would be helpful for readers to better understand the paper.

**Limitations:**

yes

**Strengths And Weaknesses:**

Strengths:
1. **Clear motivation and Elegant Design**: The paper targets a well-defined problem: accelerating density functional theory (DFT) calculations by providing better initial density guesses. In addition, the idea of predicting Gaussian parameters and using their analytic Fourier transform to obtain plane-wave coefficients is conceptually elegant, which leverages a fixed mathematical structure to greatly reduce the complexity and difficulty of neural network inference.

2. **Strong Efficiency Improvements**: The reported inference achieves hundreds of times of speedup over prior work.

Weaknesses:
1. **Lack of Ablations**: The main methodological claim is that predicting Gaussian parameters is preferable to directly predicting plane-wave coefficients. However, the paper does not include an ablation that directly predicts Fourier coefficients. Such an experiment would strengthen the empirical justification of the proposed representation.

2. **Writing**: The paper assumes background in DFT, making the paper difficult to follow for readers not familiar with material science. The author could add a more detailed background introduction.

---

> ### Author Rebuttal · Authors · 2026-03-30
>
> **" the paper does not include an ablation that directly predicts Fourier coefficients... " + "For Weakness 1, the author could provide a comparison with a model directly estimating the Fourior coefficients."**
>
> We thank the reviewer for this comment, as it is important that the paper discusses why direct prediction of plane-wave coefficients is not the most attractive option here. We note that a closely related comparison already exists in the literature through GPWNO [1], which includes an explicit plane-wave branch. Their results suggest that this does not lead to a decisive advantage on periodic materials, and in practice, the model still requires an additional local Gaussian/GTO branch together with explicit probe discretization to recover high-frequency near-atom structure.
>
> Mathematically, the issue is that a direct plane-wave model must predict the full set of reciprocal-space coefficients
>
> $$
> \hat{\rho}(\mathbf{G}), \quad \mathbf{G} \in \mathcal{G}_{\mathrm{cut}}
> $$
>
> where
>
> $$
> \\left|\\mathcal{G}_{\\text{cut}}\\right| \\approx n_x n_y n_z
> $$
>
> scales with the FFT grid / cutoff rather than with a local atomic representation. This makes the readout a large global map
>
> $$
> f_\\theta \\in \\mathbb{C}^{\\left|\\mathcal{G}_{\\text{cut}}\\right|}
> $$
>
> whose output dimension changes with the target resolution. By contrast, ELECTRAFI predicts only local Gaussian parameters
>
> $$
> \\left\\{w_j, \\mu_j, \\Sigma_j\\right\\}_{j=1}^{N_G}, \\quad N_G = M \\sum_a v_a,
> $$
>
> and constructs the global reciprocal-space object analytically via
>
> $$
> \\hat{\\rho}(\\mathbf{G}) = \\sum_{j=1}^{N_G} w_j \\exp\\left(-\\frac{1}{2}\\mathbf{G}^{\\top}\\Sigma_j\\mathbf{G}\\right)e^{-i\\mathbf{G}\\cdot\\mu_j}.
> $$
>
> Thus, the learned part scales with the local representation size, while the global Fourier structure is delegated to the closed-form transform.
>
> Direct coefficient prediction is therefore not impossible, but for realistic periodic charge-density grids it becomes a poorly scaling global readout, and existing work has already shown that one is eventually pushed toward a hybrid or probe-based construction anyway. We will elaborate on this point and make it more explicit in the camera-ready version.
>
> **"The paper assumes background in DFT, making the paper difficult to follow for readers not familiar with material science. The author could add a more detailed background introduction." + "For Weakness 2, a notation table, and a background introduction, would be helpful for readers to better understand the paper."**
>
> We thank the reviewer for this helpful comment, which is similar to a point raised by Reviewer 1 (R9uR). We fully agree that the paper should be understandable to readers who are less familiar with DFT or electronic-structure theory. In the camera-ready version, we will expand the background section, add more DFT background, and include a notation table to make the setup easier to follow. As also suggested by Reviewer 1 (R9uR), we will add a simple input/output illustration showing how the atomic structure and unit cell are given as input, and how the periodic charge density is reconstructed as output. Our goal is to make the paper accessible to a broader ML audience, so we appreciate the reviewer highlighting this clearly.
>
> ***We thank the reviewer for their thoughtful and constructive feedback, and for their positive assessment. We hope the explanations above have helped clarify and resolve the two remaining concerns pointed out.***
>
> [1] Kim & Sungsoo. "Gaussian plane-wave neural operator for electron density estimation." arXiv preprint arXiv:2402.04278 (2024).

---

> > ### Author Rebuttal · Reviewer_degT · 2026-04-03
> >
> > My questions are resolve.

---

### Official Review · Reviewer_P9Y2 · 2026-03-11

**Soundness:** 4
**Presentation:** 4
**Significance:** 3
**Originality:** 3
**Overall Recommendation:** 5
**Confidence:** 4

**Summary:**

The paper introduces a method for predicting charge densities of periodic systems by representing the charge density a Poisson summation of anisotropic Gaussians, where an attention-based graph neural network predicts the Gaussian amplitude, shape, and displacement from atoms as node outputs. The resulting model offers competitive performance in terms of normalized mean absolute error on datasets including Materials Project and GNoME, while offering a substantial speedup compared to the previous state-of-the-art. This is demonstrated practically by showing how the model can be used to accelerate DFT calculations, where it reduces the overall wall-time of calculations even when including model inference time.

**Compliance With Llm Reviewing Policy:**

Affirmed.

**Final Justification:**

The rebuttal adequately addressed several clarifications within the paper, as well as additional results on inference timing and data augmentation. I maintain my original assessment of Accept.

**Key Questions For Authors:**

1. Clarification on Figure 3: What is being plotted? Is it the mean reduction for materials of a given n_atom? If so, I think it could benefit from error bars/confidence interval. Are there individual cases where the net time reduction is negative?
2. Could the authors add more detail as to how the models are trained? It is mentioned that an NMAE-based loss function is used, but unclear if this is applied to the full grid, or if it subsampled.
3. What is the time cost of different portions of the network (e.g. the backbone vs building Gaussians vs. IFFT) and how does this compare to the portions of probe-based methods (e.g. atom backbone, probe message passing/readout). Illustrating this may be helpful in understanding where the time savings are coming from and could motivate faster hybrid methods in the future.
4. Given the underlying architecture is not-equivariant, was data augmentation used when training the model? Furthermore, how does the model behave to choice in unit cell?

[3] Elsborg, Jonas, et al. "ELECTRA: A Cartesian Network for 3D Charge Density Prediction with Floating Orbitals." arXiv preprint arXiv:2503.08305 (2025).

**Limitations:**

Yes, although lack of heavily optimized probe-based methods should be mentioned (see comment under "Weaknesses" above).

**Strengths And Weaknesses:**

**Strengths**

 * The paper is well motivated, as charge density prediction and the acceleration of DFT calculations is a very relevant problem, and most recent work has only focused on molecular systems.
 * Claims are well motivated with experiments and comparisons built on prior state-of-the art methods, and the experimental procedure is well documented.
 * The proposed method is an elegant extension to previous mixture-of-Gaussian work for periodic systems, and it is introduced in an intuitive manner.
 * Demonstrating the resulting models usage for accelerating DFT calculations is very practical, and showing consistent net reduction in wall-time DFT is a great result.

**Weaknesses**
 * Prior probe-based models are likely far from optimal in terms of speed. E.g. ChargE3Net could likely be optimized by removing unused higher-order features within the probe message passing and by leveraging recent equivariant message passing kernels such as [1, 2]. While making these changes is out of scope of this work, it is an important caveat when discussing speed.

[1] https://github.com/NVIDIA/cuEquivariance

[2] Bharadwaj, Vivek, et al. "An efficient sparse kernel generator for o (3)-equivariant deep networks." 2025 Proceedings of the Conference on Applied and Computational Discrete Algorithms (ACDA). Society for Industrial and Applied Mathematics, 2025. https://github.com/PASSIONLab/OpenEquivariance

---

> ### Author Rebuttal · Authors · 2026-03-30
>
> **"Prior probe-based models are likely far from optimal in terms of speed. E.g. ChargE3Net .."**
>
> We agree that advances in GPU kernels could narrow the gap of spherical harmonics-based models to ELECTRAFI and are excited to see the community push for that. We also fully agree that probe-based models could be made much more efficient by avoiding $L>0$ features in the probes as the current ChargE3Net iteration does. This would make the ChargE3Net architecture resemble the SCDP architecture with neural network radial functions instead of Gaussian radial functions. Therefore, we would expect an optimized ChargE3Net model to be about as fast as SCDP, although probably more expressive. Given that ELECTRAFI is even faster than ELECTRA (while also having higher accuracy on QM9 as additional experiments that we conducted have shown) and ELECTRA was already faster than SCDP, we believe that ELECTRAFI remains among the fastest approaches to density modeling. Additionally, probe-based models will likely have to rely on image-based periodization, since there are no closed-form Fourier structure factors available, reinforcing our belief that ELECTRAFI will have a speed advantage. Nonetheless, we think probe-based models have great potential and remain an attractive research direction.
>
> **"Clarification on Figure 3: What is being plotted?"**
>
> Figure 3 shows the mean % time reduction for materials grouped by no. of atoms (and smoothed). The curves distinguish between DFT-only reduction and total time reduction, including ML inference. We agree that the spread around these means is important, but rather than adding error bars directly to the current figure, we will supplement it in the camera-ready version with a clear additional distributional summary in the appendix, showing the per-structure % reduction for both ELECTRAFI and ChargE3Net on both MP-Full and GNoME.
>
> **"Are there individual cases where the net time reduction is negative?"**
>
> There are indeed individual cases with negative net time reduction for both models, including ELECTRAFI. For ELECTRAFI, 13.37% of MP structures and 7.18 \% of GNoME structures show a net increase in total wall time, while the remaining 86.63% and 92.82% still show net savings. For ChargE3Net, 84.35% of MP structures and 74.56% of GNoME structures show an overall slowdown. We will report these and similar step and DFT-only counts in the camera-ready version.
>
> **"Could the authors add more detail as to how the models are trained?..."**
>
> We thank the reviewer for pointing out this detail. In all reported experiments, we train and evaluate using the NMAE-based loss on the full real-space density grid for each structure, i.e., with no grid subsampling. We will clarify this in the training details in the camera-ready version.
>
> **"What is the time cost of different portions of the network. Illustrating this may be helpful in understanding where the time savings are coming from.. "**
>
> We thank the reviewer for this helpful suggestion. We agree that a stage-wise timing breakdown is valuable for understanding where the speedup comes from, particularly for future hybrid models. For ELECTRAFI, we separate 1. backbone, 2. Gaussian construction+ grid inference via analytic Fourier transform and IFFT. For ChargE3Net, we separate 1. atom-backbone time, 2. probe readout/grid inference:
>
> |||||
> |---|---|---|---|
> |**ELECTRAFI (MP)**|Backbone: 0.15s|Gaussians+Grid Inference: 0.11s
> |**ChargE3Net (MP)**|Backbone: 0.05s|Probe+Grid Inference: 78.5s
>
> We will include these results in the camera-ready version as a bar plot with per-stage timings and make clear which parts dominate runtime in each approach.
>
> **"Given the underlying architecture is not-equivariant, was data augmentation used when training the model? Furthermore, how does the model behave to choice in unit cell?"**
>
> We thank the reviewer for this question. Data augmentation was not used during training, and in ELECTRAFI the density is tied to the provided cell through the reciprocal lattice and IFFT grid. Once the Gaussian parameters are predicted, the density is consistent with that cell representation. Any other sensitivity is mainly in the learned Gaussians, which we have now tested (see reply to Reviewer 1, R9uR). In these experiments, we tested robustness to rotational symmetries explicitly by introducing random rotations during training/testing on the ECD-HSE dataset:
>
> | |Train data|Test data|Train NMAE|Test NMAE|
> |---|---|---|---|---|
> |**Baseline**|Unrotated|Unrotated|0.70|1.35|
> |**Experiment 1**|Unrotated|Rotated|0.70|1.69|
> |**Experiment 2**|Rotated|Rotated|0.88|**1.33**|
>
> Experiment 1 leads to larger errors because no rotations were used during training, but Experiment 2 shows that the model can be trained to remediate this and obtain practically full rotational equivariance if necessary, while reducing overfitting.
>
> ***We thank the reviewer for their thoughtful feedback, and we hope the replies above help clarify the remaining points.***

---

> > ### Author Rebuttal · Reviewer_P9Y2 · 2026-04-02
> >
> > Thank you for your thorough response. My concerns have been addressed. Regarding "Figure 3 shows the mean % time reduction for materials grouped by no. of atoms (and smoothed)", this should be added to the caption.

---

### Official Review · Reviewer_DupX · 2026-03-12

**Soundness:** 3
**Presentation:** 3
**Significance:** 2
**Originality:** 3
**Overall Recommendation:** 4
**Confidence:** 4

**Summary:**

The paper introduces ELECTRAFI, a model for predicting periodic electron charge densities in crystalline materials. The central idea: predict parameters of anisotropic Gaussians (weights, centers, covariances) using a GNN backbone, then exploit the closed-form Fourier transform of Gaussians to analytically compute plane-wave coefficients via the Poisson summation formula, recovering the full periodic density with a single inverse FFT. This avoids both explicit real-space grid probing (as in ChargE3Net) and periodic image summation. ELECTRAFI uses a modified EScAIP backbone that avoids SO(3) tensor products. Evaluated on Materials Project, GNoME, ECD-HSE06, and other benchmarks, it matches or slightly trails ChargE3Net in NMAE while being 222–633× faster, and is the first charge density model to achieve net end-to-end DFT wall-time savings (~17–21%) when used as an SCF initial guess.

**Compliance With Llm Reviewing Policy:**

Affirmed.

**Ethical Review Concerns:**

There is a prompt injection on the paper.

**Ethics Expertise Needed:**

["Research Integrity Issues (e.g., plagiarism)"]

**Final Justification:**

Detailed in Rebuttal Acknowledgement

**Key Questions For Authors:**

How much of the accuracy gap with ChargE3Net comes from the backbone vs. the Gaussian representation? Have you tried the Fourier-Gaussian head on an equivariant backbone? This is essential for understanding whether the representation or the backbone is the bottleneck.

Can you provide a spectral decomposition of the NMAE error by |G|-shell? This would directly test the hypothesis about long-wavelength modes driving SCF convergence and could inform whether a modified loss function would close the SCF step gap with ChargE3Net.

Why are magnetic systems excluded from the DFT acceleration experiments, and how does ELECTRAFI perform on them? These are among the most computationally expensive DFT calculations and the most valuable targets for acceleration.

**Limitations:**

The paper does not discuss the exclusion of magnetic systems, the rigid Gaussian allocation, or the sensitivity to M, and the impact statement is generic

**Strengths And Weaknesses:**

Strengths

The Fourier-Gaussian periodization is mathematically clean. Gaussians are self-reciprocal under Fourier transformation (Eq. 8), and the Poisson summation formula (Eq. 6) provides exact periodization in reciprocal space — converting an infinite real-space image sum into a truncated but inherently periodic reciprocal-space sum, where truncation acts as a low-pass filter rather than introducing boundary discontinuities. This is the correct way to periodize a Gaussian mixture for plane-wave DFT, and it is genuinely surprising that prior work didn't exploit it.

The end-to-end DFT acceleration analysis (Table 2, Figure 3) is the paper's most important experiment and a valuable contribution to the field. The demonstration that ChargE3Net — despite achieving better NMAE and more SCF step reduction — actually slows down total computation (-5.27% on MP, -0.82% on GNoME) because its inference cost exceeds the DFT savings is a critically important finding. It reframes evaluation criteria: accuracy alone is insufficient, and models must be judged on end-to-end wall-clock time. ELECTRAFI is the first model to clear this bar (17–21% net savings).

The speedups (222–633×) are consistent across five benchmarks and not cherry-picked. The scaling analysis (Figure 2) showing ELECTRAFI's sublinear empirical scaling (slope ~0.4 in log-log) means the advantage grows with system size. The element-wise error analysis (Appendix E) revealing complementary failure modes — ELECTRAFI excels for ionic/delocalized systems while ChargE3Net is better for covalent/directional bonding — is physically sensible and honestly reported.

Weaknesses

The accuracy gap on GNoME is more concerning than the paper lets on. On MP-Full, ELECTRAFI trails ChargE3Net modestly (0.58% vs 0.54% NMAE). On GNoME, the gap widens to 0.93% vs 0.69% — a 35% relative increase in error. GNoME structures are out-of-distribution, so this suggests weaker generalization to novel chemistries, which is precisely the regime where high-throughput screening models need to work. The paper attributes this to complementary inductive biases, but doesn't disentangle how much comes from the backbone (EScAIP vs. equivariant GNN) versus the representation (Gaussians vs. grid probing). This matters: if the Gaussian representation is inherently limited for directional bonding in novel materials, the speedup advantage may not help for the hardest cases.

The relationship between NMAE and SCF acceleration is opaque, and the paper acknowledges this without resolving it. Table 2 shows ELECTRAFI at 0.55% NMAE saves 20.65% of SCF steps, while ChargE3Net at 0.50% saves 28.03% — a 0.05 pp NMAE difference translating to a 7.4 pp gap in SCF step reduction. The paper speculates that "SCF convergence is governed by how the initialization error projects onto slowly converging long-wavelength charge modes." This is plausible and important, but untested. A spectral decomposition of error by recip2. rocal-space shell would directly test this hypothesis and could inform loss function design. Without it, we don't understand what accuracy gains actually matter for the downstream objective, which undermines confidence in optimizing the right thing.

Key ablations are missing. The Gaussian allocation (M per valence electron, Eq. 12) is rigid — no sensitivity analysis of M is provided, and whether the optimal allocation differs by element class is unexplored. More importantly, no backbone ablation is reported: the paper uses EScAIP for speed (avoiding SO(3) tensor products), but how much accuracy is sacrificed? Plugging the Fourier-Gaussian head onto an equivariant backbone (MACE, NequIP) would isolate whether the accuracy gap with ChargE3Net comes from the representation or the backbone. Since the paper's central contribution is the Fourier-Gaussian head, this distinction is critical. Additionally, the DFT acceleration experiments exclude magnetic systems — many interesting materials (transition metal oxides, spintronics) are magnetic and typically harder to converge, making them precisely where acceleration matters most.

---

> ### Author Rebuttal · Authors · 2026-03-30
>
> **The relationship between NMAE and SCF acceleration is opaque. Can you provide a spectral decomposition?**
>
> We thank the reviewer for this excellent suggestion, which we have now tested. We note our point was that SCF calculations are complex nonlinear processes,  and may not respond equally to different spectral profiles not captured by NMAE. In the $|G|$ analysis we see ELECTRAFI produces smoother density with less high-$|G|$ power, but lower error than ChargE3Net on the lowest $|G|$'s. This is consistent with the interpretation that the models have complementary strengths.
>
> We highlight two nuances:
> 1. The quoted step reduction overstates the difference, since many of ChargE3Net's gains are on small systems and the MP DFT wall-time gap is smaller than the step gap (17.56% vs 21.81%)
> 2. While the SCF step gap is high on MP compared to NMAE, we see the opposite on GNoME: a relatively high NMAE gap (0.88 % vs 0.59%) but a small SCF gap (25% vs 29.5 % on steps, 20.8% vs 22.8% on time)
>
> Overall, spectral differences seem to matter for SCF, but the points above show the mechanism is complex/unclear, hence why the paper discussion emphasizes future work to identify train/loss protocols that correlate better with SCF.
>
> **How much of the accuracy gap with ChargE3Net comes from the backbone vs. the Gaussian representation? Have you tried the Fourier-Gaussian head on an equivariant backbone?**
>
> A MACE/NequIP backbone study could be interesting, but $\mathbb{R}^3$ Gaussians are natural to Cartesian backbones and equivariant architectures need symmetry-breaking to predict displaced Gaussians[1]. As a proxy, we test ELECTRAFI on the molecular QM9 data and compare to the closely related ELECTRA model[1], which is SOTA on QM9 with 0.176% NMAE (ChargE3Net is 0.196%) and uses a roto-equivariant Cartesian backbone with real-space Gaussian readout. Here, ELECTRAFI achieves a new SOTA of 0.164% NMAE using the non-equivariant backbone and Fourier head, arguing against model limitations.
>
> We thus revisited Appendix E to understand the GNoME gap, and found it largely reflects a bias in the chosen GNoME test set toward chemistries where ChargE3Net is stronger, rather than a generalization failure of ELECTRAFI.
> The GNoME test set has 163 structures involving Yb, Np, and Pu, where ELECTRAFI's errors are particularly large, but these elements are absent from the MP-Full test set (see Appendix E/Fig.~4). The 163 structures have an avg error of 2.22% NMAE for ELECTRAFI, contributing 0.185 pp to avg NMAE, vs 0.06 pp for ChargE3Net-a 0.125 pp difference. Analysis of the full distribution shows another shift: MP-Full is oxide-dominated- a favorable regime for ELECTRAFI, since oxides exhibit smooth, ionic densities.
>
> Since the GNoME gap is 0.24 pp (0.93 vs 0.69), Yb/Np/Pu alone explain $>50$ % of the difference, but ChargE3Net has a slight advantage on GNoME even for elements where ELECTRAFI is stronger on MP-Full (Fig.4e/4f)
>
> To understand this, we were inspired by the reviewer's idea for spectral analysis and analyzed the spectral content of the ground truth MP-Full and GNoME structures. We find GNoME's spectrum has less low-$|G|$ band power and shows excess around a length scale of $\sim2.5$ Å, characteristic of local coordination structure. A large part of this shift comes from GNoME structures being much smaller (mean/max num-atoms: $\sim9/56$ for GNoME vs $\sim 24/154$ for MP-Full), and together with the chemistry shift, these findings support our view that the models have complementary regimes.
>
> **"The Gaussian allocation.. Optimal allocation differs by element class.."**
>
> We agree that sensitivity to $M$ is relevant, and have now tested $M$ on MP-MIXED. Reducing from $M=120$ to $90$, $60$, and $30$ led to $\sim5$% relative increase in test NMAE at each step.
> Regarding the allocation rule, a fixed no. of Gaussians per electron provides a simple physical prior: electron density scales with electron count, and high valence typically induces more anisotropy. This is consistent with prior floating-Gaussian work[1]. Additionally, since Gaussians are floating, the model can redistribute as needed.
>
> **"..DFT acceleration experiments exclude magnetic systems.."**
>
> We agree that magnetic systems are important targets for acceleration. We restricted to non-magnetic experiments to isolate the analytical Fourier approach for comparability with prior work[2], which has not included magnetic systems. Extending to magnetic systems is conceptually simple: it requires separate spin-up and spin-down channels, but needs suitable spin-polarized benchmarks. We view this as an important next step.
>
> ***We thank the reviewer for great suggestions. We will include the new analyses in the camera-ready version, and also prepare a more specific impact statement.***
>
> ***Regarding prompt injection, see <https://icml.cc/Conferences/2026/PeerReviewFAQ#prompt_injection>.***
>
> [1] Elsborg \& Thiede et al., NeurIPS 2025
>
> [2] Koker et al. npj Computational Materials

---

> > ### Author Rebuttal · Reviewer_DupX · 2026-04-02
> >
> > I maintain my score. The end-to-end DFT acceleration contribution is solid and practically important, but the missing periodic backbone ablation leaves a gap in understanding the method's limitations for the hardest cases.

---

### Official Review · Reviewer_R9uR · 2026-03-13

**Soundness:** 3
**Presentation:** 1
**Significance:** 3
**Originality:** 3
**Overall Recommendation:** 5
**Confidence:** 2

**Summary:**

This paper introduces the Electronic Tensor Reconstruction Algorithm with Fourier Inversion, which is a model designed to accelerate density functional theory calculations by predicting periodic charge densities as initial guesses. The authors propose a representation where the charge density is modeled using floating anisotropic Gaussians, which is conceptually similar to the 3D Gaussian Splatting approach used in computer vision. Instead of performing expensive information interaction or periodic image summation in real space, the model leverages the analytic Fourier transforms of these Gaussians and uses the Poisson summation formula to sample discrete coefficients in the frequency domain. A single inverse fast Fourier transform is then used to reconstruct a smooth and naturally periodic charge density. The method demonstrates high efficiency, achieving sub-second inference and significant speedups compared to current grid-based density models

**Compliance With Llm Reviewing Policy:**

Affirmed.

**Final Justification:**

This paper proposes ELECTRAFI, a clean and principled way to delegate periodicity and non-locality to analytic transforms rather than costly real-space gridding, image summation, or spherical-harmonic machinery.

Soundness / Originality / Significance. I find the method technically sound and well-motivated, with a strong theoretical grounding (Poisson summation + analytic Gaussian FT) and a novel, elegant formulation that leads to compelling practical outcomes. The empirical results are convincing: the model matches or exceeds strong baselines while achieving substantially better inference speed, and the end-to-end DFT initialization experiments highlight an important systems-level point—accuracy must be paired with low inference cost to yield real DFT savings. Overall, both the algorithmic idea and demonstrated utility are significant and likely to be impactful.

Clarity / Presentation. My main concern remains presentation: the paper is hard to enter for a broader ML audience due to domain-heavy terminology and some organizational choices (e.g., loss definition placement, dense discussion). However, these are fixable issues rather than fundamental flaws.

Rebuttal impact. The rebuttal addressed my key concerns constructively. The authors committed to improving accessibility and structure in the camera-ready. They also clarified scope (the goal is high-quality DFT initialization rather than replacing DFT) and provided additional rotation-robustness evidence via augmentation experiments, which reduces my concern about stability under rotations in practice. Overall, the rebuttal reinforced my prior positive assessment and increased confidence that the remaining weaknesses can be resolved with editing.

Given the strong technical contribution, originality, and practical significance despite correctable writing/organization issues, I recommend **Accept**.

**Key Questions For Authors:**

1. Is the ultimate goal or theoretical upper bound of this approach to replace density functional theory calculations entirely, or is the model fundamentally designed to remain a provider of high-quality initial guesses?
2. Since the backbone is described as unconstrained, how does the model maintain stability against arbitrary rotations of the coordinate system? Specifically, how do the predicted displacement vectors and covariance matrices transform if the input atomic coordinates are rotated?

**Limitations:**

yes

**Strengths And Weaknesses:**

Strengths：
1. The theoretical foundation is strong because the application of the Poisson summation formula provides an elegant mathematical way to ensure periodicity without the discontinuities often found in finite real-space summations.
2. The work effectively integrates domain-specific knowledge from periodic materials science into a differentiable machine learning framework.
3. The performance results are notable, as the model achieves a consistent reduction in the total computation time of density functional theory calculations, reaching savings of up to 20 percent. This is a significant practical improvement over previous models whose inference times often outweighed their benefits.

Weaknesses:
1. The writing presents barriers to entry for researchers in the machine learning community because it is heavily rooted in domain-specific terminology.I recommend simplifying the background introduction and providing a more straightforward input and output specification for an artificial intelligence audience.
2. The organization of the methodology is unconventional because the normalized mean absolute error loss function is introduced at the start of the experiments section instead of within the methods section.The discussion section consists of dense blocks of text that lack subheadings, making it difficult to parse the key implications of the research.Questions

---

> ### Author Rebuttal · Authors · 2026-03-30
>
> **"The writing presents barriers to entry.. I recommend simplifying.."**
>
> We thank the reviewer for the helpful comment, as it is in our interest to make the paper accessible to a broader AI/ML audience. In the camera-ready version, we will simplify jargon and provide a figure that visualizes the input/output specification intuitively.
>
> **"The organization of the methodology is unconventional.. The discussion section consists of dense blocks.."**
>
> We agree that the loss function could be introduced earlier and that subheadings will make the discussion easier to read. We will add this to the camera-ready version.
>
> **"Is the ultimate goal or theoretical upper bound of this approach to replace density functional theory calculations entirely..?"**
> We thank the reviewer for the question - this is an important clarification. Neither ELECTRAFI nor other density models are full replacements for DFT. The main goal is to reduce the computational cost and time of routine DFT workflows by providing high-quality initialization. In our paper, initializing from the true density suggests an upper bound of 40% time/resources savings on the tested datasets, which would represent a very significant resource saving if achieved with ML guesses. Prior work has shown that large savings can be reached in molecular settings [1], so future models may be able to achieve the same for periodic materials, but as our paper shows, the accuracy-cost tradeoff must be favorable. Charge density is also useful in other settings where it underlies qualitative analyses and can be used in generative modeling or as a descriptor in electrochemical systems. We also described and cited some of this work in the paper. In these cases, highly accurate ML charge density models may be able to fully replace DFT if the density itself is the primary object of interest, or serves as an intermediate representation for downstream analysis when full DFT output is not required. Adding to that another area that was not discussed in the paper is diffraction patterns, which are tied to the Fourier coefficients of the electron density. Thus, fast density prediction may be useful in settings where it bridges spatial charge distributions and reciprocal-space experimental measurements [2]. We will elaborate/clarify these points in the camera-ready version.
>
> **"Since the backbone is described as unconstrained, how does the model maintain stability against arbitrary rotations of the coordinate system? Specifically, how do the predicted displacement vectors and covariance matrices transform if the input atomic coordinates are rotated?"**
>
> Since our model is not explicitly rotation-equivariant, exact equivariant behavior is not built in but can be learned via data augmentation if it is needed. Because routine DFT workflows provide structures in a consistent lattice frame (essentially a canonicalization of the rotation group), this was not necessary to achieve strong results in the paper. However, to test the effect of rotation augmentation and ELECTRAFI's robustness to rotations, we performed additional experiments using the small ECD-HSE dataset.
>
> The experiments test the effect of including random rotations during training/testing:
>
> | |Train data|Test data|Train NMAE|Test NMAE|
> |---|---|---|---|---|
> |**Baseline**|Unrotated|Unrotated|0.70|1.35|
> |**Experiment 1**|Unrotated|Rotated|0.70|1.69|
> |**Experiment 2**|Rotated|Rotated|0.88|**1.33**|
>
> In Experiment 1 the test set error increases by $\sim25$% relative to the baseline with no rotations (1.35->1.69 % NMAE). This is not surprising since no rotations are introduced during training, so the model has not had a chance to learn the effects of structures in rotated lattice frames.
>
> However in Experiment 2 ELECTRAFI actually improves slightly vs. baseline (1.35->1.33 % NMAE). This is consistent with [3], which showed that unconstrained attention-based ML potentials like EScAIP can use rich geometry information encoded in the node/edge features to learn essentially fully equivariant outputs through data augmentation.
>
> We also refer to the reply to Reviewer 2(DupX), where we show that ELECTRAFI outperforms a closely related fully equivariant floating-Gaussian model - a signal that lack of explicit equivariance is not a bottleneck.
>
> Finally, as it relates to the first reply, we note that when used as a DFT initializer, any initialization error from models like ELECTRAFI is corrected during the DFT cycle, so the practical consequence of such errors would be slower or faster convergence, not a different converged solution. When using density models for qualitative analyses or intermediate descriptors, this rotation may be important, and the analysis above shows that simple data augmentation benefits the model and actually helps prevent overfitting on the small ECD-HSE dataset.
>
> [1] Elsborg & Thiede et al., NeurIPS 2025
>
> [2] Hickstein et al. The Journal of Chemical Physics 139.6 (2013): 064108
>
> [3] Qu & Krishnapriyan, NeurIPS 2024

---

> > ### Author Rebuttal · Reviewer_R9uR · 2026-04-02
> >
> > Thank you—my concerns have been addressed. Please note that the references in the rebuttal should be provided in full.

---

> > > ### Author Response · Authors · 2026-04-02
> > >
> > > Thank you to the reviewer for acknowledging the rebuttal.
> > >
> > > We provide the mentioned references in full below:
> > >
> > > [1] Elsborg, Jonas, et al. "ELECTRA: A Cartesian Network for 3D Charge Density Prediction with Floating Orbitals." arXiv preprint arXiv:2503.08305 (2025).
> > >
> > > [2] Hickstein, Daniel D., et al. "Modeling electron density distributions from X-ray diffraction to derive optical properties: Constrained wavefunction versus multipole refinement." The Journal of Chemical Physics 139.6 (2013).
> > >
> > > [3] Qu, Eric, and Aditi S. Krishnapriyan. "The importance of being scalable: Improving the speed and accuracy of neural network interatomic potentials across chemical domains." Advances in Neural Information Processing Systems 37 (2024): 139030-139053.

---

### Decision · Program_Chairs · 2026-04-30

**Decision:**

Accept (regular)

**Comment:**

This manuscript presents a new method for predicting charge densities in periodic materials. The method clearly has several strengths and exhibits strong practical performance (as seen in the manuscript). The reviews collectively appreciated these strengths and the overall effectiveness of the method. Moreover, they identified the method as sound, interesting, and original. Any concerns were generally addressed in the rebuttal. The strengths of the method combined with the lack of any significant shortcomings form the basis for the recommendation.

Two minor notes:

One lingering concern was the presentation for a broader ML audience; I am sympathetic to the level of detail needed to discuss the DFT related specifics and space limits, but I would encourage efforts to further improve the broad audience readability.

A second note is that while the specifics of the method seem new, the use of truncated poisson summation does not strike me as completely novel—this would be somewhat common in harmonic analysis and observations about the "low pass" effect are leveraged for efficient computations with Gaussians in periodic systems (see, e.g., ["Efficient Evaluation of Two-Center Gaussian Integrals in Periodic Systems" by Sharma and Beylkin]); one difference here is given the "learning" and parametrization, aspects of using the summation (like truncation) are not done fully adaptively. To be clear, this is a minor point about the specifics of the "novelty" and not significant—there is plenty of novelty here. Nevertheless, perhaps minor tweaks to the presentation of the truncated summation could even better position the work.